# MEMENTO: TOWARD AN ALL-DAY PROACTIVE ASSISTANT FOR ULTRA-LONG STREAMING VIDEO

**Hongxiang Jiang**[1]  **Zengrui Ge**[2]  **Guo Chen**[3]  **Qixiong Wang**[4]
**Jile Jiao**[1]  **Xuetao Feng**[1]  **Yuan Wang**[5]  **Yan Wang\***[1]
[1]Deepeleph Intelligent Technology   [2]Henan Univeristy
[3]Beijing University of Aeronautics and Astronautics   [4]Xiaohongshu   [5]Tsinghua University
`wy84378@intime.com.cn`

## ABSTRACT

Multimodal large language models have demonstrated impressive capabilities in visual-language understanding, particularly in offline video tasks. More recently, the emergence of online video modeling has introduced early forms of active interaction. However, existing models, typically limited to tens of minutes, are not yet capable of all-day proactive understanding over ultra-long video streams. They struggle to maintain long-term context online, as they suffer from token accumulation and lack scalable memory mechanisms. These limitations hinder critical tasks such as reminding users that medication was taken hours earlier—an ability that exemplifies the shift from reactive to memory-oriented assistants with long-term reasoning. To bridge this gap, we present Memento, the first proactive vision-language framework for ultra-long streaming video. To avoid token growth and support scalable long-duration understanding, we introduce Dynamic Memory and Query-related Memory Selection, enabling sparse memory retention and efficient retrieval. To address the training challenges of memory-based modeling, we propose Step-Aware Memory Attention, which aligns memory access with temporal steps for stable supervision. To support both training and evaluation of active, long-term behavior, we construct Memento-54K and MementoBench, a dataset-benchmark suite covering diverse tasks on text, object, and action across video streams up to 7 hours. Experiments demonstrate that Memento achieves superior performance, paving the way toward reliable all-day proactive video assistants.

## 1  INTRODUCTION

Recent advancements in large language models (LLMs) (Ouyang et al., 2022; Touvron et al., 2023; Yang et al., 2024b;a; Xin et al., 2025; Guo et al., 2025) and vision-language models (VLMs) (Liu et al., 2023; 2024a; Achiam et al., 2023; Bai et al., 2025) have shown remarkable progress in video understanding, particularly with the emergence of long-form (Ren et al., 2024; Song et al., 2024a; Zeng et al., 2025) and online video LLMs (Chen et al., 2024a; Wu et al., 2024b; Li et al., 2025a; Qian et al., 2025). Such progress has further raised expectations for an all-day, proactive assistant. This assistant would continuously perceive the environment through ultra-long video streams and proactively interact with humans, rather than merely responding passively to explicit user queries. Achieving this capability would not only fundamentally transform the role of AI assistants in daily human activities, but also represent a critical step toward genuine autonomous agents (Fan et al., 2024; Wang et al., 2024b; Putta et al., 2024; Hong et al., 2024).

Despite this promising progress, existing models still fall short of realizing such a proactive assistant in practice. Their limitations become especially evident in scenarios requiring extremely long-term behavioral monitoring and temporal reasoning. For instance, an all-day assistant should be able to recall whether the user has already taken a specific medication hours ago, detect that the same object has been accessed multiple times throughout the day, or notice a warning text previously ignored. Fig. 1 illustrates a detailed case, inspired by a scene from the film *Memento*: the wife asks for an insulin shot three times within a few hours, but the husband, due to short-term memory loss, fails to recognize the repeated requests, potentially leading to serious consequences. In this scenario, existing long-form video models fail to assist during critical moments. They cannot issue timely warnings during the shots and fail to respond accurately. On the other hand, even the most advanced online

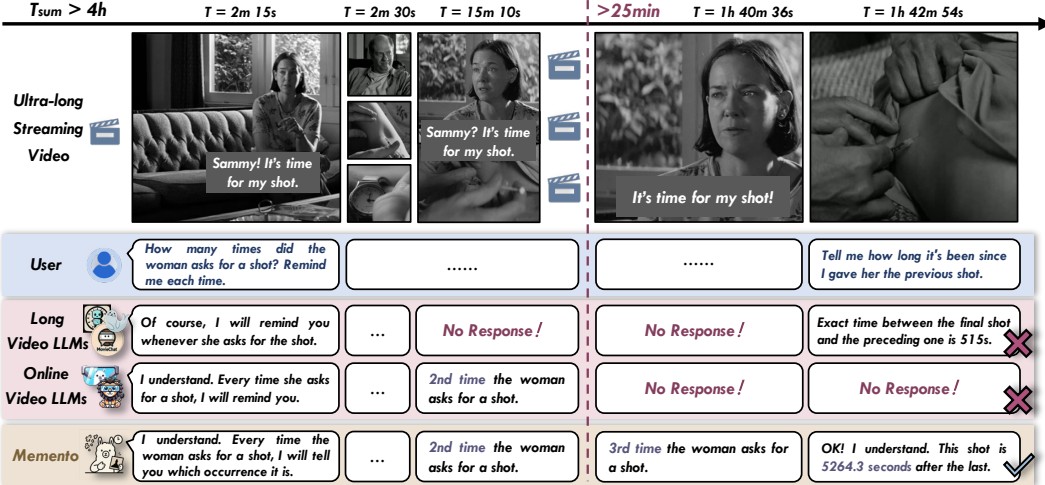

Figure 1: **Comparison of model behaviors for all-day proactive assistance.** Long-term and online video models both fail to assistant at injection points beyond 25 minutes. Conversely, Memento continuously tracks repeated shots, demonstrating its capability toward serving as an all-day proactive assistant. Results for online models and Memento are obtained via supervised fine-tuning (SFT), while long video model outputs are based on prompt engineering due to architectural limitation.

streaming models struggle with ultra-long durations due to their token-based architectures, which cause visual tokens from each frame to accumulate in memory usage over time. As a result, after at most a few dozen minutes, the model exceeds GPU memory limits, and cannot recognize that previous requests occurred.

To address the above issues, we propose **Memento**, a proactive vision-language framework for ultra-long video streams. To handle the long-term memory challenge, we introduce a Dynamic Memory (DM) mechanism that learns to retain or fuse incoming visual information over time, allowing Memento to preserve relevant context while keeping memory usage bounded. In addition, we propose a Query-related Memory Selection (QMS) module that retrieves only the most relevant memory slots during generation, enabling efficient and targeted access across extended video durations. This framework departs from the token-based paradigm, in which frame-level features are concatenated and multiple positions are supervised jointly. In contrast, Memento operates over dynamically updated memory representations, which evolve over time and cannot be aligned to discrete frame steps. As a result, directly applying token-level supervision leads to misaligned inputs and invalid training. To resolve this structural mismatch, we introduce Step-Aware Memory Attention (SAMA), which restricts attention to memory available at each step, ensuring temporally consistent and semantically valid learning. While Memento addresses the architectural challenges of proactive interaction with long-range memory, existing datasets (Chen et al., 2024a; Yao et al., 2025; Grauman et al., 2022; Yang et al., 2025) offer limited support for training or evaluation. Online benchmarks (Chen et al., 2024a; Li et al., 2025b; Wu et al., 2024a) include only short-term proactive tasks such as behavior recap based on recent frames, lacking supervision for long-term monitoring. To bridge this gap, we construct **Memento-54k** and **MementoBench**, covering diverse task types on text, object, and action over video streams up to 7 hours, all requiring long-range, proactive understanding.

Our contributions are summarized as follows:

- **Framework.** For the first time, a framework for proactive interaction over ultra-long video streams, named Memento, is proposed.
- **Memory modeling.** To address the scalable long-term memory challenges, we introduce dynamic memory and a query-related selection for selective retention and efficient retrieval.
- **Training strategy.** To enable training compatibility with dynamic memory, we propose step-aware memory attention, ensuring stable and effective learning for proactive vision-language modeling.
- **Dataset and benchmark.** We construct Memento-54k and MementoBench, covering diverse long-range proactive tasks, validating the effectiveness of Memento and supporting the development of an all-day proactive assistant.

| Related Work | Visual Input | Long Form | Proactive |
|---|---|:---:|:---:|
| LLaMA-VID (ECCV 2024) | *fixed token* | ✓ | ✗ |
| TimeSuite (ICLR 2025) | *fixed token* | ✓ | ✗ |
| MovieChat (CVPR 2024) | *fixed memory* | ✓ | ✗ |
| MA-LMM (CVPR 2024) | *fixed memory* | ✓ | ✗ |
| VideoLLM-online (CVPR 2024) | *fixed token* | ✗ | ✓ |
| VideoLLM-MoD (NeurIPS 2024) | *dynamic token* | ✗ | ✓ |
| LION-FS (CVPR 2025) | *dynamic token* | ✗ | ✓ |
| **Memento** | ***dynamic memory*** | ✓ | ✓ |

Table 1: **Comparison between related methods and the proposed Memento.** "Proactive" indicates whether the model supports interaction without explicit queries.

## 2  RELATED WORK

**Long-Form Video Understanding.** Recent multimodal large language models have demonstrated strong instruction-following capabilities in video understanding (Cheng et al., 2024; Zhang et al., 2023; Li et al., 2024b; Liu et al., 2024b; Wang et al., 2025; Zhang et al., 2024b), particularly for long-range content. As early approaches based on sparse frame sampling often fail to capture key clues in long videos (Lin et al., 2024; Li et al., 2024a; Maaz et al., 2024; Ma et al., 2024; Zhou et al., 2024), fixed token-based methods have been introduced via encoding each frame into a fixed number of visual tokens, with compression algorithm for acquiring more frames (Wang et al., 2024c; Ren et al., 2024; Weng et al., 2024). For example, LLaMA-VID (Li et al., 2024c) represents each frame only using two visual tokens, enabling efficient processing of hour-long videos. Beyond token-based compression, fixed memory-based models, including MovieChat (Song et al., 2024b), Koala (Tan et al., 2024), MA-LMM (He et al., 2024), and others (Fan et al., 2024; Wang et al., 2024b; Zhang et al., 2024a), maintain a fixed-length memory bank as the visual tokens. They achieve effective long-video compression by aggregating redundant frames with similar features. However, these approaches suffer from increasing inference overhead, limited long-term memory and the inability to proactively interact, making them unsuitable for all-day assistant scenarios.

**Online Video LLMs.** Online Video LLMs aim to achieve real-time, proactive interaction over streaming inputs, with the ultimate ambition of supporting continuous operation across ultra-long video streams in open-ended scenarios. VideoLLM-online (Chen et al., 2024a) is the first to enable proactive interaction in video-language modeling by introducing a Streaming-EOS objective to decide when to respond or remain silent. However, like other fixed token-based approaches, it requires extracting visual tokens for each incoming video frame, leading to unacceptable growth in memory usage and computational cost. To reduce overhead, subsequent models introduce dynamic token strategies, such as MoE-style (Jacobs et al., 1991; Fedus et al., 2022; Shazeer et al., 2017; Lepikhin et al., 2021) token routing in VideoLLM-MoD (Wu et al., 2024b) and LION-FS (Li et al., 2025a), where only a subset of tokens are forwarded into deeper layers, and patch-level token dropping in TimeChat-online (Yao et al., 2025), where high redundancy regions are discarded. These methods increase the supported video duration to tens of minutes, but still retain frame token accumulation. Even the most advanced multimodal models (Wang et al., 2024a; Chen et al., 2024c; Gao et al., 2024; Chen et al., 2024b), such as GPT-4o (Achiam et al., 2023) and Gemini 1.5 Pro (Team et al., 2024; 2023), struggle to proactively reason over ultra-long streaming video. Unlike prior works, Memento introduces a dynamic memory design and query-related retrieval, as shown in Table. 1, avoiding token burden and preserving relevant information beyond fixed memory limits. Overall, it paves the way toward reliable, all-day proactive assistants..

## 3  MEMENTO: A PROACTIVE LLM OVER ULTRA-LONG VIDEO STREAMS

### 3.1  OVERVIEW

In this section, we introduce our Memento in detail. As shown in Fig. 2 (a), given a streaming video $\mathcal{V} = \{f_1, f_2, \ldots, f_T\}$, Memento encodes each frame $f_t$ using a ViT-based (Radford et al., 2021) encoder. The result $v_t \in \mathbb{R}^{(1+h_p \times w_p) \times C}$ contains a global [CLS] token and $h_p \times w_p$ spatial tokens.

Instead of directly projecting $v_t$ into the language space via an MLP projector as in LLaVA (Liu et al., 2023; 2024a), we first process it through the Dynamic Memory (DM). At each step $t$, the current $v_t$ and historical memory $\mathcal{M}_{t-1}$ are fused according to a Remember-and-Forget (R&F) strategy. It decides whether to retain the original information, and produces the updated memory

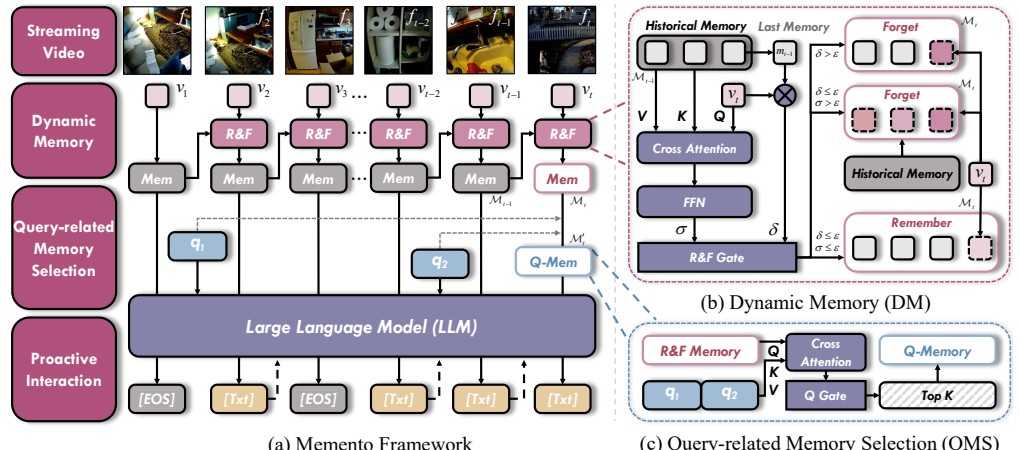

(a) Memento Framework

(b) Dynamic Memory (DM)

(c) Query-related Memory Selection (QMS)

Figure 2: **Overall architecture of Memento.** (a) Memento receives user queries and historical responses with the current memory state, achieving proactive interaction over ultra-long video streams. (b) Details of the DM mechanism, which mainly utilizes similarity-based retention and aggregation. (c) Details of the QMS module using query-conditioned gating and masking.

$\mathcal{M}_t = \mathrm{DM}(v_t, \mathcal{M}_{t-1})$ as *R&F Memory*. Then, for all the user queries $q = \{q_1, q_2, \ldots, q_n\}$ in the past, the current R&F memory $\mathcal{M}_t$ is filtered by the Query-related Memory Selection (QMS) to retrieve the most relevant subset $\mathcal{M}'_t \subset \mathcal{M}_t$. The selected memory $\mathcal{M}'_t$ is fed into the LLM to generate the next-token distributions $P$, enabling both reactive and proactive responses.

Finally, considering that the fused memory changes across frames but lacks per-frame token structure, we apply Step-Aware Memory Attention (SAMA) to restrict attention to available memory at each time step during training. Thanks to this alignment, the supervision objective from VideoLLM-online (Chen et al., 2024a) can be directly adopted to train the memory-based framework:

$$\mathcal{L} = \frac{1}{N} \sum_{j=1}^{N} (\underbrace{- \log l_{j+1} P_j^{[\mathrm{Txt}_{j+1}]}}_{LM\ Loss} - \underbrace{\log f_j P_j^{[\mathrm{EOS}]}}_{Streaming\ Loss}), \tag{1}$$

where $l_j$ is 1 if the $j$-th token is a language response token, and 0 otherwise. $f_j$ is 1 if both (1) the $j$-th token is the last token in $\mathcal{M}'_t$, and (2) $l_{j+1}$ is 0. $P_j^{[\mathrm{Txt}_{j+1}]}$ is the probability on the $j + 1$-th text token, output from the large language model head of the $j$-th token, and $P_j^{[\mathrm{EOS}]}$ is the probability for the EOS token.

## 3.2 DYNAMIC MEMORY

To update R&F memory $\mathcal{M}_t$, we aim to balance between retaining essential information and fusing redundant content, which may arise in the short term (adjacent frames with little change) or in the long term (repeated scenes or actions), as shown in Fig. 2 (b). To handle both, we compute two relevance scores: (1) a short-term score $\delta$, based on cosine similarity (Wang et al., 2024d;e) between the current frame $v_t$ and the last memory $m_{t-1} \in \mathbb{R}^{(1+h_p \times w_p) \times C}$ in $\mathcal{M}_{t-1}$; and (2) a long-term score $\sigma$, obtained via cross-attention (Vaswani et al., 2017) between $v_t$ and all flattened historical memory tokens $M_{t-1} \in \mathbb{R}^{N_{t-1}(1+h_p \times w_p) \times C}$ in Eq. 2. A fixed threshold $\epsilon$ controls memory update.

$$\mathrm{Attn}(v_t, \mathbf{M}_{t-1}) = \mathrm{softmax}\left(\frac{(v_t W_q)(M_{t-1} W_k)^\top}{\sqrt{d}}\right),$$
$$\sigma = \psi\left((\mathrm{Attn}(v_t, M_{t-1}) \cdot (M_{t-1} W_v)) W_o\right), \tag{2}$$

where $W_q$, $W_k$, $W_v$, and $W_o$ are projection matrices. $\psi(\cdot)$ denotes a summation followed by a sigmoid activation (LeCun et al., 1998), yielding a scalar score $\sigma \in \mathbb{R}$.

The R&F gate selects the memory update strategy based on a relevance threshold $\epsilon$. If $\delta > \epsilon$, the current frame is considered locally redundant and is fused into the last memory token using Eq. 3.

$$\mathrm{score} = \mathrm{softmax}(\mathrm{Attn}(m_{t-1}, v_t)), \quad w = \mathrm{score} \cdot u,$$
$$\tilde{m}_{t-1} = m_{t-1} \cdot (1 - \mathrm{sum}(w)) + w^\top v_t, \quad \mathcal{M}_t = \mathrm{Concat}(\mathcal{M}_{t-1}^{[:N_{t-1}-1]}, \tilde{m}_{t-1}), \tag{3}$$

where score $\in \mathbb{R}^{(1+h_p \times w_p) \times (1+h_p \times w_p)}$ is the normalized attention weight in spatial, $u \in \mathbb{R}$ is a fixed scalar update ratio. $\tilde{m}_{t-1}$ is the fused token, and $\mathcal{M}_{t-1}^{[:N_{t-1}-1]}$ denotes the first $N_{t-1} - 1$ memory. If $\delta \le \epsilon$ while $\sigma > \epsilon$, the frame is semantically aligned with long-term memory content; we thus reuse the same update strategy but compute attention scores by treating $M_{t-1}$ as queries and $v_t$ as keys and values, enabling soft updates across all memory slots. Finally, if both $\delta \le \epsilon$ and $\sigma \le \epsilon$, the frame is considered distinct and directly appended to memory, namely $\mathcal{M}_t = \text{Concat}(\mathcal{M}_{t-1}, v_t)$.

This gated update mechanism enables Memento to forget redundant content via token fusion, and remember distinct information. Different from token-based methods, this mechanism could avoid unacceptable growth in memory usage and computational cost. Compared with the fixed-length memory banks, it dynamically expands for novel content. This design maintains a compact yet expressive representation across ultra-long video streams.

### 3.3 QUERY-RELATED MEMORY SELECTION

To reduce memory consumption while preserving response quality, we filter the current R&F memory $\mathcal{M}_t$ according to user queries $q$ in Fig. 2 (c). Specifically, we transform $\mathcal{M}_t$ into $M_t \in \mathbb{R}^{N_t \times (1+h_p \times w_p) \times C}$, and compute cross-attention with user tokens $Q$ as keys and values, following Eq. 2, to yield the score $R \in \mathbb{R}^{N_t}$ for each memory frame. QMS then applies a top-$k$ gating strategy to select the most relevant $k = r_{\text{qms}} \cdot N_t$ tokens, $\mathcal{M}_t' = \text{TopK}(M_t, R, k)$. The selected compact memory $\mathcal{M}_t'$ is then passed to the LLM for generation. Our QMS ensures query-aware generation while decreasing the cost of full-memory attention, thereby enabling scalable reasoning over ultra-long temporal sequences.

### 3.4 STEP-AWARE MEMORY ATTENTION

Unlike token-based models with frame-wise accumulation, the memory bank lacks explicit alignment with video steps. Thus, prior standard training methods in (Chen et al., 2024a; Wu et al., 2024b; Li et al., 2025a) with causal attention are inapplicable. As shown in Fig. 3 (a), this attention will allow access to expired memory. In contrast, our proposed SAMA in Fig. 3 (b) introduces a masking scheme to align with frame-wise visibility.

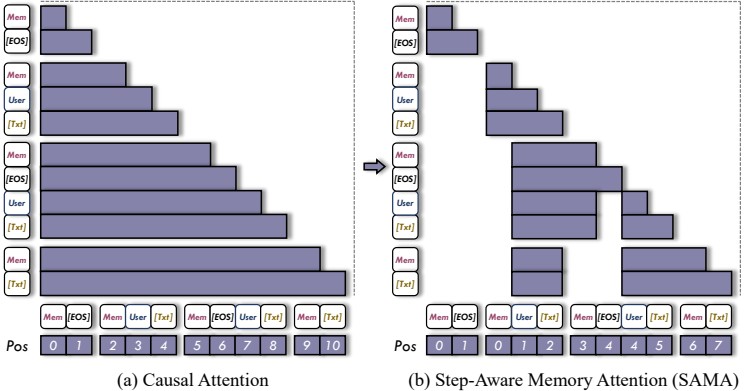

Figure 3: **Causal attention vs. SAMA.** Causal attention (left) permits access to all past tokens, including expired memory. SAMA (right) restricts attention to valid memory, excluding irrelevant tokens.

Specifically, an example of input sequence is:

$$\text{tokens} = [\mathcal{M}_1', [\text{EOS}], \mathcal{M}_2', q_1, [\text{Txt}]_1, \mathcal{M}_3', [\text{EOS}], q_2, [\text{Txt}]_2, \mathcal{M}_4', [\text{Txt}]_3]_L. \quad (4)$$

A binary attention mask $A \in \{0, 1\}^{L \times L}$ is built, where token $x_i$ is allowed to attend to token $x_j$ if:

$$A_{ij} = \begin{cases} 1, & x_j \in \mathcal{M}_s' \cup q \cup \{[\text{Txt}]_k\}_{k=1,2,..}, \ i \ge j, \ x_j \ne [\text{EOS}] \\ 1, & i = j, \ x_i = [\text{EOS}] \\ 0, & \text{otherwise} \end{cases} \quad (5)$$

Here, $s = \text{step}(x_i)$ denotes the video frame index when token $x_i$ is added to the sequence. Furthermore, we reassign correct position ids for each token to ensure that tokens within the same frame share a base offset. This aligns positional encoding with the token visibility defined by the mask.

During inference, we maintain the same masking structure so that only previous dialog tokens are stored as key-value cache (Dao et al., 2023; Ge et al., 2024), allowing efficient streaming decoding with minimal computation.

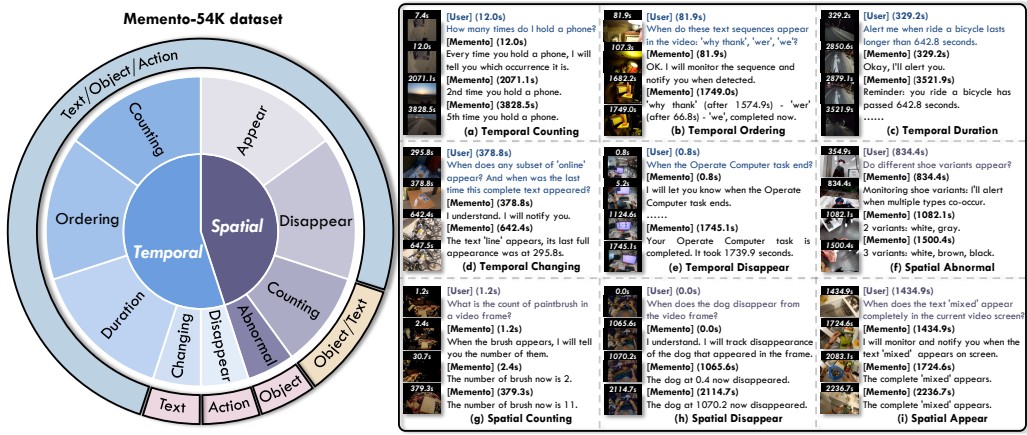

Figure 4: **Overview of Memento-54k.** Left: the 9 task types categorized by spatial vs. temporal, and by modality (text, object, action). Right: example QA instances for each task type.

# 4  DATASET AND BENCHMARK: MEMENTO-54K AND MEMENTOBENCH

## 4.1  MEMENTO-54K DATASET CONSTRUCTION

**Video Filtering and Sampling**. To support long-duration, proactive interaction, we construct Memento-54k based on Ego4D (Grauman et al., 2022). We filter all videos to retain those between 5 minutes and 7 hours, to ensure long-term context. To reduce sample imbalance, we downsample videos from overrepresented scenarios (e.g., cooking), yielding a subset of 4,466 daily-life videos.

**Task Annotation**. As illustrated in Fig. 4, we define 9 task types spanning spatial and temporal reasoning, where spatial tasks focus on short-term perception (e.g., object presence), and temporal tasks require long-range memory (e.g., repeated actions or text changing). These tasks are designed for three modalities: action, object, and text. Each sample is annotated as a streaming QA pair, including a question and multiple assistant responses with timestamps. For each modality, we first obtain timestamp-level labels, and then generate QA pairs:

- *Action.* Based on Ego4D timestamp narrations, we prompt GPT-4o to generate QA pairs such as repeated actions of Temporal Counting on the right side of Fig. 4, see Appendix for details.
- *Object.* We extract objects at 2 FPS using ChatReX (Jiang et al., 2024), a category-agnostic detector. QA pairs are then generated via rule-based scripts. For example, in temporal duration tasks, we track object appearance and disappearance timestamps to identify presence for producing response.
- *Text.* On-screen text is detected by Qwen2-VL at 2 FPS. Text annotations are similar to object, such as temporal changing tasks identify cases where a previously seen full text is later partially disappeared, and once the subset is matched, a response is triggered to form a QA pair.

**Streaming QA Formatting.** For each task, up to 9 instances are annotated per video, each focusing on a distinct action, object, or text. Failed or invalid cases are manually corrected. Then, QA pairs are grouped by randomly selecting 1-5 user queries with their timestamped responses to form new streaming samples. This forms the final release of the Memento-54k dataset, and the specific distribution is as shown in Table. 2.

| Split | Duration | Videos | Samples | Responses |
|---|---|---|---|---|
| Train | Total | 4,426 | 53.6k | 2.5M |
| Test | 5-10 min | 10 | 62 | 2.6k |
| | 10-30 min | 10 | 67 | 3.6k |
| | 30-60 min | 15 | 56 | 4.8k |
| | > 60 min | 5 | 13 | 2.5k |
| | Total | 40 | 198 | 13.5k |

Table 2: **Distribution of Memento-54k.**

Especially, streaming QA must scan entire videos, making evaluation expensive. Though it contains only 40 videos, the test set covers over 13k responses, which is sufficient for robust evaluation.

## 4.2  MEMENTOBENCH EVALUATION

To evaluate models under the proactive long-term understanding setting, we identify three essential requirements for this task: temporal alignment, answer quality, and minimal redundancy.

***TimeRecall***. *TimeRecall* is the fraction of ground-truth responses for which the model produces at least one response within a 5-second window, reflecting the ability to anticipate when to respond.

***Score***. *Score* measures the generation quality by comparing all the model responses within the above window with ground-truth answers. We use GPT-3.5-turbo-0125 to assign score from 1 to 10 and take the maximum among multiple outputs. Scoring details are provided in the Appendix.

***Redundancy***. *Redundancy* captures the extent of unnecessary generation, defined as the proportion of model responses outside the time window in *TimeRecall*.

The most closely related benchmark to ours is Ego4D Narration Stream (Chen et al., 2024a; Lin et al., 2022), which evaluates the temporally align performance on generated descriptions with visual events in streaming egocentric videos. However, it focuses only on the current narration, overlooking tasks that require long-term past information. In addition, its evaluation relies on exact text match, whereas MementoBench supports free-form outputs, enabling more flexible and robust assessment.

Notably, existing online benchmarks (Li et al., 2025b; Wu et al., 2024a) such as OVO-Bench, which appear to evaluate proactive interaction, in fact offer an offline-form question and predefined response timestamps during inference. All past video frames before each timestamp are provided, which ideally should be judged by the model. As a result, such benchmarks emphasize response accuracy for specified questions, allowing non-proactive models to be evaluated under this setting. In contrast, MementoBench compares whether models can proactively interact at the right time with the correct content, enabling more accurate evaluation of the desired capabilities in real-world proactive settings.

## 5 EXPERIMENTS

### 5.1 IMPLEMENTATION DETAILS

In this work, we implement our Memento following the VideoLLM-online framework (Chen et al., 2024a). Unless otherwise stated, we use SigLIP-ViT-L/384 (Zhai et al., 2023) as the vision encoder, which extracts frame-wise features at 2 FPS, and set $h_p = w_p = 3$. For the LLM module, we use LLaMA-3.1-8B-Instruct (Grattafiori et al., 2024). Following (Chen et al., 2024a), we train 1 epoch for our model in the DeepSpeed Zero-2 (Rajbhandari et al., 2020) configuration, with LoRA (Hu et al., 2022) to all linear layers in the LLM with a rank of 128 and a scaling factor of 256. For our DM module, we set the relevance threshold $\epsilon = 0.7$, and update ratio $u = 0.2$. In the QMS module, the top-$k$ ratio $r_{qms} = 50\%$. We use AdamW optimizer (Loshchilov & Hutter, 2019) with a learning rate of $1e\text{-}4$ and cosine decay. All experiments are conducted on 4 NVIDIA A100 GPUs (80GB). Please refer to Appendix for inference details with a dynamic correction strategy.

### 5.2 MAIN RESULTS

We compare our method with VideoLLM-online (Chen et al., 2024a) using MementoBench. To ensure fairness, we train VideoLLM-online with our Memento-54k dataset using the same training schedule, denoted as VideoLLM-online*.

To assess runtime scalability, Figure 5 (right) shows GPU memory usage during streaming video inference. VideoLLM-online quickly accumulates tokens and runs into OOM at about 25 minutes, with memory peaking at 80.5 GB. In contrast, Memento maintains bounded usage under 45.3 GB across the entire 4-hour streaming videos, demonstrating its advantages for proactive response to ultra-long videos with stable memory and no interruption. The occasional rises correspond to dense response periods and are reduced afterward as temporary variables are released.

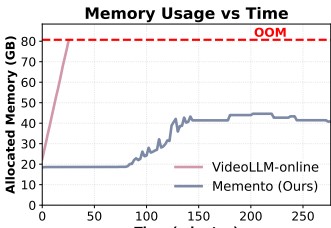

Figure 5: **Memory Usage.**

The results on MementoBench are shown in Table 3. The original VideoLLM-online performs poorly across all aspects, with only 6.1% spatial and 11.8% temporal recall, and nearly 0% beyond 25 minutes due to memory overflow. Even after supervised fine-tuning (SFT) on Memento-54k (VideoLLM-online*), average recall only rises to 8.9%, with long-term recall still at 0.3%. While it reports a higher score of 5.32 and lower redundancy of 21.3%, this is largely because it triggers

| Method | TimeRecall ↑ | | | | Score ↑ | | | Redund. ↓ |
|---|---|---|---|---|---|---|---|---|
| | Sp. | Temp. | Long (> 25min) | Avg. | Sp. | Temp. | Avg. | |
| Online Video LLMs | | | | | | | | |
| VideoLLM-online | 6.1% | 11.8% | 0.1% | 8.1% | 1.55 | 1.21 | 1.40 | 56.4% |
| VideoLLM-online* | 7.9% | 11.6% | 0.3% | 8.9% | 5.11 | 5.68 | 5.32 | 21.3% |
| Ours | | | | | | | | |
| Memento* | **45.9%** | **51.3%** | **35.2%** | **47.5%** | 4.31 | 4.02 | 4.22 | 64.5% |

Table 3: **Evaluation on MementoBench.** *Sp.* and *Temp.* denote spatial and temporal task types, where *Temp.* requires long-term visual reasoning. *Long* marks responses beyond 25 minutes, for assessing understanding persistence under ultra-long video streams, independent of task type. In particular, VideoLLM-online is the only model with available open-source online inference code.

| Memory Schema | TimeRecall ↑ | | | Score ↑ | | | Redund. ↓ |
|---|---|---|---|---|---|---|---|
| | Sp. | Temp. | Avg. | Sp. | Temp. | Avg. | |
| Fixed Memory | | | | | | | |
| Len=8 | 14.8% | 22.1% | 16.9% | 4.61 | 4.65 | 4.64 | 55.5% |
| Len=32 | 20.4% | 26.4% | 22.1% | 5.14 | 5.04 | 5.12 | 53.7% |
| Len=128 | 28.1% | 31.2% | 29.0% | 4.77 | 4.74 | 4.76 | 52.7% |
| Dynamic Memory | | | | | | | |
| $\epsilon=0.6$ | 23.1% | 25.5% | 23.8% | **5.12** | **5.25** | **5.16** | **50.9%** |
| $\epsilon=0.7$ | 38.2% | **46.7%** | 40.4% | 4.36 | 4.67 | 4.39 | 56.2% |
| $\epsilon=0.8$ | **43.9%** | 46.6% | **44.7%** | 4.59 | 4.05 | 4.43 | 61.4% |

Table 4: **Ablation on memory schema.** "Len" indicates the fixed memory bank size. Our dynamic memory consistently yields better recall and offers superior trade-offs in others.

Figure 6: **Memory Size Comparison.**

very few responses, often staying silent when answers are expected. The resulting low recall makes it unsuitable for real-world applications. In contrast, Memento achieves 45.9% spatial, 51.3% temporal, and 35.2% long-duration recall, while maintaining a solid score of 4.22. Although its redundancy increases to 64.5%, given the substantial gain in recall (+38.6%), we consider this a worthwhile trade-off, as ensuring timely and consistent response is critical in ultra-long online scenarios.

## 5.3 ABLATION STUDY

We conduct three ablation studies to evaluate the core design components of Memento. Our analysis focuses on three aspects: memory mechanism (with $1+2\times2$ frame tokens, $r_{qms}=100\%$), frame token configuration (with $\epsilon=0.7$, $r_{qms}=100\%$) and QMS top-k ratio (with $\epsilon=0.7$, $1+2\times2$ frame tokens).

**Memory Mechanism.** To examine the impact of memory structure and hyperparameter on long-term reasoning, we compare fixed-length memory banks with our dynamic memory mechanism, as shown in Table 4. Increasing the fixed memory size improves recall from 16.9% to 29.0% and slightly reduces redundancy. In comparison, dynamic memory achieves notably higher recall (up to 44.7% at $\epsilon=0.8$) while maintaining comparable score and redundancy (up to 5.16 and 50.9% at $\epsilon=0.6$). Notably, for temporal tasks that require long-range memory, recall improves significantly from 31.2% (fixed) to 46.7% at $\epsilon=0.7$. Figure 6 further shows that dynamic memory scales naturally with video length, enabling long-range context retention. However, $\epsilon=0.8$ results in nearly $10\times$ larger memory than $\epsilon=0.7$ with marginal gain in all the metrics, so we adopt the default $\epsilon=0.7$.

**Frame Token Configuration.** We futher analyze different frame tokens in Table 5. $1+3\times3$ offers a better balance, which achieves the highest recall of 68.9%, while maintaining a reasonable score of 3.78 and moderate redundancy at 66.6%. Fewer tokens achieve a too low recall of 40.4%. $1+4\times4$ increases redundancy without improving recall.

| Frame Token | TimeRecall ↑ | Score ↑ | Redund. ↓ |
|---|---|---|---|
| $1+2\times2$ | 40.4% | **4.39** | **56.2%** |
| $1+3\times3$ | **68.9%** | 3.78 | 66.6% |
| $1+4\times4$ | 60.9% | 3.93 | 67.2% |

Table 5: **Ablation on frame tokens.**

| $r_{qms}$ | TimeRecall ↑ | | | Score ↑ | | | Redund. ↓ | Memory Usage ↓ |
|---|---|---|---|---|---|---|---|---|
| | Sp. | Temp. | Avg. | Sp. | Temp. | Avg. | | |
| 10% | 38.7% | 31.5% | 33.6% | 4.31 | **4.67** | 4.33 | 63.0% | 39.53 GB |
| 50% | **54.7%** | **59.5%** | **56.1%** | 3.96 | 3.86 | 3.93 | 66.7% | 45.19 GB |
| 90% | 49.0% | 52.8% | 50.1% | 4.15 | 3.97 | 4.10 | 63.7% | 53.50 GB |
| 100% | 38.2% | 46.7% | 40.4% | **4.36** | **4.67** | **4.39** | **56.2%** | 55.44 GB |

Table 6: **Ablation on QMS top-$k$ ratio.** We exclude textual KV cache in "Memory Usage" reporting, as dialogue history size varies with response behavior and is independent of $r_{qms}$.

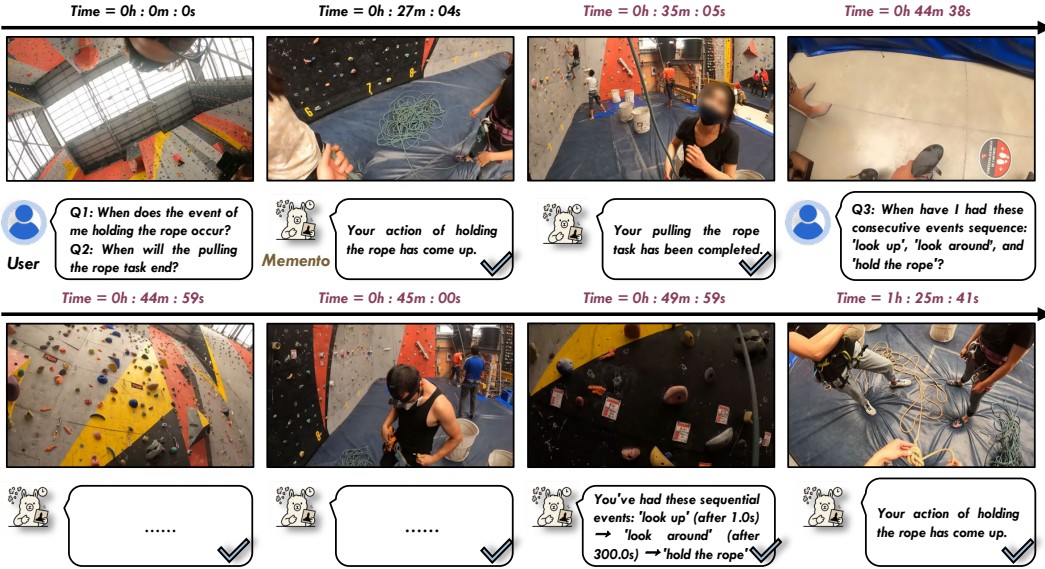

Figure 7: **Qualitative results of Memento on ultra-long streaming video.** The scene involves rock climbing over a 1.5-hour timeline, with three user queries issued at 0, 0, and 44 minutes, respectively. These queries cover the tasks of spatial appear, temporal disappear and temporal ordering for action.

**QMS Top-$k$ Ratio.** To assess how QMS filtering affects retrieval relevance, we adjust the top-$k$ selection ratio $r_{qms}$ in the QMS module. As shown in Table 6, selecting all memory slots ($r_{qms}$=100%) results in suboptimal performance: although it achieves the highest score of 4.39 and lowest redundancy of 56.2%, its recall is notably lower compared to the best $r_{qms} = 50\%$ setting by 15.7%. This highlights that overly broad memory access may introduce irrelevant context and distract attention from key visual evidence. Meanwhile, too few slots ($r$=10%) limits context recall and harms performance. The 50% configuration strikes the best trade-off across all metrics, demonstrating that QMS effectively prioritizes relevant memory and improves response alignment.

## 5.4 VISUALIZATION OF MEMENTO

Figure 7 showcases Memento's performance on a 1.5-hour streaming video with temporally distant queries. The model identifies "holding the rope" at 27 minutes in response to an initial query and triggers "pulling the rope completed" 8 minutes later. It also tracks the ordered occurrence of "look up", "look around", and "hold the rope" before issuing a final response. Moreover, it remains proactive across the entire duration, generating correct responses even after 80 minutes, demonstrating its robustness in ultra-long streaming scenarios.

## 6 CONCLUSION

In this paper, we present Memento, a proactive vision-language framework for ultra-long streaming video. It introduces dynamic memory, query-related selection and step-aware attention for scalable long-term context modeling and temporally aligned training. Moreover, we construct Memento-54k and MementoBench for training and evaluation. Experiments show that Memento enables effective proactive interaction. Declaration of LLM usage will be discussed in Appendix.

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
