# A  APPENDIX

## CONTENTS

## A.1 DETAILED BEHAVIOR OF SAMA MODULE

The Step-Aware Memory Attention (SAMA) module defines how attention and position encodings are assigned for memory-based streaming video modeling, as illustrated in Fig. 1. Specifically:

- Only the current memory is retained, and all earlier memory tokens are discarded.
- All previous user queries and assistant responses remain visible to support long-term reasoning.
- Any [EOS] token is masked out from future attention—no later token can see it.

These constraints ensure that meaningful interactions remain persistently accessible, while non-response markers such as [EOS] are excluded to prevent memory accumulation and interference during decoding. To preserve temporal consistency, position IDs are propagated from the most recent remembered dialogue turn, enabling the model to build upon prior context. If no memory is retained, position IDs are instead reinitialized to avoid drift from irrelevant history.

```
offset = 0
# (1) Causal attention and local position ids
for block in causal_tokens:
    start, end = block_range(block)
    attention_mask[start:end, start:end] = tril(1)
    position_ids[start:end] = range(0, end - start)
# (2) Global offset shift for remembered QA (always-attend dialog)
for start, end in remembered_QA_tokens:
    attention_mask[end:, start:end] = 1
    position_ids[end:] += position_ids[end - 1] + 1 - offset
    offset = position_ids[end - 1] + 1
# (3) Mask out [EOS] and reset position alignment
for start, end in eos_tokens:
    attention_mask[end:, start:end] = 0
    if end < seq_len and tokens[end] is not Memory:
        # shift to align with last remembered content
        position_ids[end:] -= end - start
```

Figure 1: Algorithmic illustration of attention masking and position encoding in SAMA.

**Position logic.** In implementation, SAMA updates position IDs based on the latest remembered dialog (e.g., from a past user turn). The position ID of the first token in the current memory continues from the end of that remembered span. If a token follows a masked [EOS], it continues from the most recent valid memory, skipping over the [EOS] as if it never existed in the context timeline.

## A.2 INFERENCE DETAILS

To balance the timing of model responses in streaming video, VideoLLM-online introduces a correction strategy. Specifically, if the predicted probability of [EOS] falls below a fixed threshold $\theta$, it is forced to zero; otherwise, it remains silent. However, we observe this approach to be highly sensitive in practice, even minor changes in training configuration may cause over-response in one case and under-response in another. This makes consistent evaluation difficult and requires expensive manual tuning per model. To address this, we design a simple dynamic adjustment strategy during inference, from an inverse perspective: instead of suppressing [EOS], we explicitly require the probability of generating a [Txt] token to exceed a threshold $\theta'$ before triggering a response. Then, the threshold $\theta'$ will be initialized at 0.5 and increased by $\Delta_{\theta'}=0.1$ after 10 consecutive response frames to reduce over-generation. If no responses occur for 30 frames, $\theta'$ resets. This adaptive mechanism stabilizes behavior across models and is consistently applied in all our evaluations.

## A.3 PERFORMANCE ON OVBENCH BENCHMARK

Beyond our long-form and proactive setting, we further evaluate our Memento on established online benchmarks for assessing its generalization. To this end, we conducted additional experiments on OVBench (Huang et al., 2025), a recently proposed benchmark for streaming vision-language understanding, covering diverse online tasks, as shown in Table 7.

| Method | AVG | FP | | | THV | | | PM | | | SP | | STP | | TP | | |
|---|---|---|---|---|---|---|---|---|---|---|---|---|---|---|---|---|---|
| | | AA | GSP | MP | AP | SV | OP | AR | PR | TR | AL | OP | AT | OT | AS | SL | OES |
| ★ VideoChat-Online | **54.9** | 64.1 | 59.7 | 16.6 | 63.1 | 58.3 | **62.8** | **42.2** | 54.4 | 70.6 | **54.1** | 24.8 | 88.7 | 48.5 | 73.0 | 25.9 | **71.7** |
| VideoChat-Online | 53.9 | 56.4 | 63.0 | 15.6 | 57.1 | 57.9 | 61.9 | 39.1 | 54.2 | **73.9** | 41.3 | 29.7 | 92.2 | **53.1** | 69.8 | 27.3 | 69.9 |
| Gemini-1.5-Flash | 50.7 | **71.4** | 53.6 | 21.9 | 56.5 | **60.8** | 40.6 | 36.7 | 47.9 | 62.5 | 32.3 | 37.5 | 87.0 | 50.0 | **83.3** | 22.3 | 46.9 |
| Qwen2-VL | 49.7 | 60.3 | **66.1** | 22.1 | 54.9 | 51.5 | 51.1 | 37.8 | 64.4 | 69.3 | 35.3 | 28.5 | 97.0 | 49.4 | 65.1 | 30.8 | 11.7 |
| LLaVA-OneVision | 49.5 | 68.0 | 62.7 | **35.9** | 58.4 | 50.3 | 46.5 | 29.4 | 60.7 | 58.0 | 43.1 | 14.2 | 86.5 | 49.7 | 70.7 | 28.1 | 30.2 |
| InternVL2-7B | 48.7 | 52.6 | 60.2 | 27.6 | 57.5 | 52.0 | 58.5 | 38.8 | **67.1** | 58.3 | 38.1 | 31.3 | 87.4 | 37.0 | 75.4 | 31.4 | 5.9 |
| InternVL2-4B | 44.1 | 57.7 | 57.0 | 14.4 | 59.2 | 49.4 | 60.0 | 30.3 | 61.8 | 46.3 | 30.9 | 20.1 | 83.0 | 32.3 | 70.7 | 29.4 | 3.4 |
| LongVA | 43.6 | 64.1 | 56.5 | 29.5 | 54.9 | 51.9 | 34.8 | 35.3 | 55.6 | 57.7 | 31.6 | 3.4 | 67.4 | 44.7 | 80.0 | 26.7 | 4.0 |
| LLaMA-VID | 41.9 | 43.6 | 50.9 | 19.6 | 64.0 | 47.5 | 46.8 | 29.4 | 48.9 | 51.2 | 31.9 | 11.2 | 75.7 | 24.8 | 59.1 | 26.0 | 40.0 |
| MiniCPM-V 2.6 | 39.1 | 33.3 | 35.9 | 15.0 | 59.2 | 50.8 | 55.1 | 25.0 | 37.4 | 41.7 | 26.6 | 11.8 | **98.3** | 36.3 | 66.1 | 26.4 | 6.2 |
| VTimeLLM | 33.1 | 37.2 | 23.4 | 15.0 | **64.8** | 43.8 | 53.2 | 25.9 | 38.8 | 32.5 | 25.9 | 20.4 | 40.9 | 6.8 | 48.4 | **43.5** | 8.6 |
| ★ Flash-Vstream | 31.2 | 26.9 | 37.6 | 23.9 | 60.1 | 41.9 | 40.0 | 23.4 | 35.3 | 26.1 | 24.7 | 28.8 | 27.0 | 21.4 | 29.8 | 25.6 | 26.8 |
| ★ MovieChat | 30.9 | 23.1 | 27.5 | 23.6 | 58.4 | 43.9 | 40.3 | 25.6 | 31.1 | 23.9 | 26.9 | 39.6 | 24.4 | 28.9 | 29.3 | 25.5 | 21.9 |
| LITA | 20.4 | 19.2 | 24.5 | 19.9 | 40.8 | 48.9 | 24.9 | 3.1 | 27.3 | 6.4 | 6.9 | 14.6 | 35.2 | 23.9 | 27.4 | 0.5 | 3.4 |
| TimeChat | 12.8 | 7.7 | 15.3 | 18.7 | 20.6 | 15.7 | 11.7 | 9.1 | 14.7 | 9.8 | 7.5 | 19.5 | 13.9 | 10.3 | 9.3 | 10.1 | 10.8 |
| ★ VideoLLM-Online | 9.6 | 0.0 | 1.8 | 20.9 | 5.2 | 5.9 | 32.6 | 0.0 | 2.3 | 26.7 | 0.6 | 26.6 | 0.9 | 19.9 | 0.9 | 1.7 | 8.3 |
| ★ **Memento (Ours)** | 48.5 | 48.7 | 59.9 | 35.6 | 57.9 | 53.7 | 60.5 | 32.2 | 57.6 | 57.4 | 36.3 | **40.1** | 64.8 | 36.5 | 64.9 | 36.8 | 33.6 |

Table 7: **Comparison on OVBench.** ★ indicates the input is streaming video. *FP* (Future Prediction) includes *AA* (Action Anticipation), *GSP* (Goal/Step Prediction) and *MP* (Movement Prediction). *THV* (Temporal Hallucination Verification) includes *AP* (Action Persistence), *SV* (Step Verification) and *OP* (Object Presence). *PM* (Past Memory) includes *AR* (Action Retrieval), *PR* (Procedure Recall), and *TR* (Trajectory Retrieval). *SP* (Spatio Perception) includes *AL* (Action Location) and *OP* (Object Position). **STP** (Spatio-Temporal Perception) includes *AT* (Action Trajectory) and *OT* (Object Trajectory). **TP** (Temporal Perception) includes *AS* (Action Sequence), *SL* (Step Localization) and *OES* (Object Existence State).

| Method | Training | Testing | TimeRecall ↑ | | | | Score ↑ | | | Redund. ↓ |
|---|---|---|---|---|---|---|---|---|---|---|
| | | | Sp. | Temp. | Long (> 25min) | Avg. | Sp. | Temp. | Avg. | |
| **Memento+ (Ours)** | w/o A + B | A | 33.3% | 30.1% | 22.0% | 31.1% | 4.90 | 5.01 | 4.96 | 35.8% |
| **Memento+ (Ours)** | w/o A + B | B | 28.7% | 34.9% | 26.9% | 31.6% | 4.43 | 3.67 | 4.10 | 57.0% |

Table 8: **Zero-shot evaluation on MementoBench.** A corresponds to *Crafting / Knitting / Sewing / Drawing / Painting* scenario, B corresponds to *Cooking* scenario.

OVBench includes 16 tasks for streaming visual-language understanding. To ensure fair comparison, we trained our model using a subset of the VideoChat-Online training data, specifically VideoChat-Online-1T and 0.27M samples from VideoChat2-1T (approximately 1/7 of the full dataset used by some advanced methods). Despite using only a fraction of the training data compared to larger baselines, our method achieves strong performance, outperforming long-context models such as LongVA and MovieChat, and matching the performance of InternVL2. Notably, our model surpasses online understanding methods like Flash-VStream and VideoLLM-online. While we fall short of VideoChat-Online with 6.4%, we attribute this gap largely to training scale. Most importantly, we emphasize that this experiment serves to demonstrate compatibility and generalization. Our primary goal remains enabling proactive assistance in ultra-long video streams, the capability not captured by existing online benchmarks. In fact, neither the tasks nor the baselines in OVBench are designed to measure such behavior.

## A.4 ZERO-SHOT EVALUATION ON MEMENTOBENCH

Regarding zero-shot generalization performance, due to the unique nature of proactive streaming assistance over ultra-long videos, no directly compatible benchmark currently exists, so we design a strict zero-shot setting within the Memento-54k dataset. Specifically, we removed the two most frequent scene categories of "Crafting/knitting/sewing/drawing/painting" and "Cooking" (labeled by original Ego4D dataset) from training, reducing the training set size from 53.6k to 42.6k samples, and tested exclusively on these unseen scenarios. The results is shown in Table 8 This reduces the training data by 21% with an expected performance drop. Nevertheless, Memento retained stable long-range proactive response behavior and competitive scores. This experiment demonstrates that its core proactive capabilities can extend, to a certain degree, beyond the training scenes.

## A.5 MEMENTO-54K ANNOTATION

### A.5.1 ACTION MODALITY

Our action-oriented annotation comprises six types spanning short-term perception and long-term temporal understanding. Based on timestamped narrations from Ego4D, we employ GPT-4o to automatically construct 37,024 fine-grained question-answer (QA) pairs for action modality.

**Action Spatial Appear.** This task aims to identify the frame where an instance of a countable user action becomes visually observable in the video. The prompt for data generation is detailed below.

---

**The Prompt for Spatial Appear Task in Action QA Generation**

You are an excellent expert in understanding long video descriptions. Please follow the instructions below and, based on the provided video captions, help me label the data:

─────────────────────────

Please strictly follow these requirements for annotation:
1. In the provided video captions, the pronoun **"you"** refers to the user (i.e., "I").
2. In assistant responses, **"you"** refers to the user.
3. Automatically identify all countable events in the video captions.
- **An event is countable if it refers to an action or occurrence that can be quantified.**
4. For each identified event, generate the following in JSON format:
   • A user question inserted at the timestamp of the first occurrence minus 1 second (use 0 if result is negative), asking about the timing of the event.
   • For each occurrence of the event, insert an assistant response indicating that the user's action has occurred.
5. The final output must conform to the following JSON format:

```
[{
    "event": "Event Name",
    "data": [{
        "user": "User question related to identifying the event occurrences.",
        "time": Insertion time of the question (seconds)
    },{
        "assistant": "Assistant acknowledgment response.",
        "time": Insertion time of the question (seconds)
    },{
        "assistant": "Your action of [event] has come up.",
        "time": Time when the first occurrence of the event happens (seconds)
    }, ...
    ]}, ...
]
```

──────── Reference Video Descriptions ────────

---

**Action Spatial Disappear.** Following the spatial appear task, the spatial disappear task aims to detect the point at which an individual action instance becomes no longer visually observable. GPT-4o is prompted to identify such moments and return the end timestamp for each individual occurrence.

---

**The Prompt for Spatial Disappear Task in Action QA Generation**

You are an excellent expert in understanding long video descriptions. Please follow the instructions below and, based on the provided video captions with timestamps, help me label the data:

─────────────────────────

**Annotation Guidelines:**
1. In the provided video captions, the pronoun **"you"** refers to the user (i.e., "I").
2. When generating user questions, replace **"you"** with **"I"**.
3. In assistant responses, **"you"** refers to the user.
4. **Identify Events with Timestamps:**
   • Extract all events from the video captions along with their corresponding timestamps.
   • Ensure that the events are listed in chronological order.
5. **Select Relevant Events:**
   • Identify events that occur two or more times in the video.
   • Select up to **three** such events.
6. **Determine Event End Timestamps:**
   • For each occurrence, the end timestamp is the time of the next event that is different.
   • If the event is the last one, its end timestamp equals its start timestamp.
   • If the same event occurs back-to-back, treat each as a separate occurrence.
7. **Format the Output:**
   • Insert a user question at the beginning (time = 0) asking when the event ends.
   • Insert an assistant acknowledgment response at the same timestamp.
   • For each occurrence, add an assistant response stating that the event has ended.
8. The final output must strictly conform to the following JSON format: ...

──────── Reference Video Descriptions ────────

---

**Action Temporal Duration.** It focuses on estimating the duration of a single narrated event as it continuously occurs in the video. GPT-4o is prompted to determine when each event starts and ends.

---

### The Prompt for Temporal Duration Task in Action QA Generation

You are an excellent expert in understanding long video descriptions. Please follow the instructions below and, based on the provided video captions with timestamps, help me label the data:

___________________________

**Annotation Guidelines:**
1. In the provided video captions, the pronoun **"you"** refers to the user (i.e., "I").
2. When generating user questions, replace **"you"** with **"I"**.
3. In assistant responses, **"you"** refers to the user.
4. **Identify Events with Timestamps:**
   • Extract all events from the video captions with their corresponding timestamps.
   • Ensure events are ordered chronologically.
5. **Select Relevant Events:**
   • Identify events that occur **two or more times**.
   • Select up to **three** such events.
   • Event names must match their wording in the video captions.
6. **Determine Start and End Timestamps:**
   • For each occurrence, define the **start** as the current timestamp.
   • The **end** is the timestamp of the next different event.
   • If the event is last in sequence, its end equals its start.
   • Back-to-back identical events are treated as separate occurrences.
7. **Calculate Duration:**
   • For each occurrence, compute the duration as `end - start` in seconds.
8. **Format the Output:**
   • At the beginning, insert:
     – A user question at `time = 0`: `"How long did each [event] last?"`
     – An assistant acknowledgment at `time = 0`: `"I will inform you of the duration each time [event] ends."`
   • For each event occurrence, add:
     – `start` and `end` timestamps
     – An assistant message indicating when the event ended and how long it lasted
     – A `time` field showing the duration in seconds
9. The final output must strictly conform to the following JSON format: ...

———— Reference Video Descriptions ————

---

**Action Temporal Disappear.** This task focuses on identifying when a high-level action disappears, as indicated by a semantically coherent sequence of events bounded by a clear starting and ending event. Since the ending event alone is insufficient, the model must reason over prior context to determine whether the action has concluded. This task evaluates temporal abstraction and context-aware understanding in streaming long-form videos.

---

### The Prompt for Temporal Disappear Task in Action QA Generation

You are an expert in analyzing video captions to identify tasks with distinct start and end events, and calculating the duration of these tasks. Please follow the instructions below to help me label the data based on the provided video captions.

___________________________

**Annotation Guidelines:**
1. **Identify Tasks with Distinct Start and End Events:**
   • Identify tasks in the video captions.
   • Each task must have a **unique start event** and a **distinct end event**.
   • Select up to **three** tasks from the video.
   • Use the **exact wording** of the start and end events as described in the captions.
2. **Calculate Task Durations:**
   • For each task, compute the time difference between the start and end event timestamps.
3. **Generate Data for Each Task:**
   • At the task's start time (or `time = 0` for the first task), insert:
     – A user question: `"When will the [task] task end?"`
     – An assistant response: `"I will let you know when the [task] task ends."`
   • At the task's end time, insert:
     – A response indicating task completion and total duration: `"Your [task] task has been completed. It took [duration] seconds."`
4. **Format the Output:**
   • The final output should be a JSON array with entries for each task.
   • Each entry includes the task name, start/end event names, and a list of time-stamped user/assistant interactions.
5. The output must strictly follow the structure below: ...

———— Reference Video Descriptions ————

---

**Action Temporal Counting.** It focuses on tracking and responding to repeated occurrences of identifiable actions in a video. Below is the prompt used to generate the annotations.

---

**The Prompt for Temporal Counting Task in Action QA Generation**

You are an excellent expert in understanding long video descriptions. Please follow the instructions below and, based on the provided video captions, help me label the data:

————————————————————

**Annotation Guidelines:**
**Important Notes:**
- In the provided video captions, the pronoun **"you"** refers to the user (`"I"`).
- Replace **"you"** with **"I"** when generating user questions.
- In assistant responses, **"you"** refers to the user.
1. **Identify Countable Events:**
   - Detect all countable events from the video captions.
   - An event is countable if it refers to an action or occurrence that can be clearly quantified.
2. **Select Relevant Events:**
   - Only include events that occur **two or more times**.
   - Select up to **three** such events.
3. **Generate Data for Each Event:**
   - Insert a user question and assistant acknowledgment **one second before** the first occurrence (or at `time = 0` if the result is negative):
     – User: `"When does the event of me [event] occur?"`
     – Assistant: `"I understand. Every time you [event], I will remind you."`
   - For each subsequent occurrence:
     – Add: Assistant: `"You have [event]."`
     – Include the corresponding timestamp.
4. **Format the Output:**
   - The final output is a JSON array, one entry per event.
   - Each entry includes the event name and a sequence of time-stamped interactions.
5. The output must strictly follow the structure below: ...

——————— Reference Video Descriptions ———————

---

**Action Temporal Ordering.** It emphasizes the recognition of consecutive event sequences that may span long temporal intervals, and requires the assistant to respond upon their completion. It evaluates the model's ability to capture long-range temporal dependencies and track consistent action orderings.

---

**The Prompt for Temporal Ordering Task in Action QA Generation**

You are an excellent expert in understanding long video descriptions. Please follow the instructions below and, based on the provided video captions, help me label the data:

————————————————————

**Annotation Guidelines:**
**Important Notes:**
- In the provided video captions, the pronoun **"you"** refers to the user (`"I"`).
- Replace **"you"** with **"I"** when generating user questions.
- In assistant responses, **"you"** refers to the user.
1. **Identify All Events and Their Timestamps:**
   - Extract all events along with their corresponding timestamps.
   - Only consider events with **identical wording** as the same event.
   - Sort events chronologically.
2. **Identify the Two Most Frequent Event Pairs:**
   - A valid event pair is formed when **two different events occur consecutively** with no other events in between.
   - Identify the **two most frequent** such pairs across the captions.
3. **Generate Data for Each Event Pair:**
   - At `time = 0`, insert:
     – User: `"When did I have these consecutive events of [event 1] and then [event 2]?"`
     – Assistant: `"I understand. Every time you have these consecutive events, I will remind you."`
   - For each valid pair occurrence:
     – Insert: Assistant: `"The [event 1] and then [event 2] has occurred."`
     – Include both event timestamps and set the message time to the second event's timestamp.
4. **Format the Output:**
   - The final output should be a JSON array with two entries—one per frequent event pair.
   - Each entry includes the pair description, timestamps, and user/assistant dialogue.
5. The output must strictly follow the structure below: ...

——————— Reference Video Descriptions ———————

---

### A.5.2 OBJECT MODALITY

In this section, we describe our seven object-oriented tasks for generating QA pairs based on dense object-level narrations, extracted using ChatReX at 2 FPS. Each narration includes a timestamped list of objects, such as *"<69, 278, 659, 539><wooden cabinet>"*. Based on these structured annotations, we generate initial 74,742 QA pairs by applying a fixed user-assistant interaction pattern, where placeholders are filled in accordingly. All QA pairs are stored in structured JSON format before further human correction.

Below is an example of the predefined QA pattern used for the Temporal Ordering Task:

- **User:** *"When do these object sequences appear in the video: '[A]', '[B]', '[C]'?"*
- **Assistant:** *"I understand. I will monitor the object sequence: '[A]' → '[B]' → '[C]' and notify you when detected."*
- **Assistant (at detection time):** *"Object sequence detected: '[A]' (after [X]s) → '[B]' (after [Y]s) → '[C]', completed at [T]s."*

**Object Spatial Appear.** Its goal is to identify when a specific object appears in the video frame. Objects are temporally counted along the video timeline, and only those with a moderate frequency of occurrence are retained for QA generation.

**Object Spatial Disappear**. Its goal is to identify when a specific object disappears from the video frame. We compute the maximum continuous visible duration for each object and retain only those whose presence lasts sufficiently long.

**Object Spatial Counting.** Its goal is to identify how many instances of a specific object appear in the current video frame. For robustness, we retain objects with a moderate number of total appearances, which are based on total frame count, with defaults set to [10, 30] for short videos.

**Object Temporal Duration.** It evaluates a model's temporal sensitivity by detecting both the duration of continuous visibility and the duration of absence between object reappearances. We retain objects with long visible or invisible durations and a moderate number of total appearances.

**Object Temporal Counting.** Its goal is to count how many times a specific object appears throughout the video. We retain objects with a moderate number of total appearances.

**Object Temporal Ordering.** Its goal is to detect when a specific sequence of three distinct objects appears in a consistent temporal order. We identify valid triplets that occur repeatedly (e.g., 7–20 times) with proper temporal spacing, and track their appearance timings across the video.

**Object Temporal Abnormal.** Its goal is to identify co-occurrence of multiple variants of the same base object (e.g., different colors or sizes of "cup") within a single frame. We retain base objects that exhibit 2 or more distinct variants co-occurring at 2–10 distinct time points.

### A.5.3 TEXT MODALITY

In this section, we describe our seven text-oriented tasks for generating QA pairs based on dense timestamped OCR results, extracted using Qwen2.5-VL at 2 FPS. Each frame-level annotation provides a list of visible text strings, such as *"Welcome to the Museum"*. After normalization and filtering for different languages, we generate initial 40,447 QA pairs by applying a fixed user-assistant interaction pattern, where placeholders are filled with the selected text content. All QA pairs are stored in structured JSON format before further human refinement.

**Text Spatial Appear.** Its goal is to identify when a specific text appears completely in the video frame. We retain text spans that are semantically complete and appear with moderate frequency (e.g., 7-20 times).

**Text Spatial Disappear.** Its goal is to detect when a specific text starts to leave the video screen. We retain text spans with complete structure and disappearance durations exceeding 30 seconds.

**Text Spatial Counting.** Its goal is to count how many instances of a specific text appear in each video frame. We keep text blocks that appear in multiple frames, and at least one frame contains multiple instances of this text.

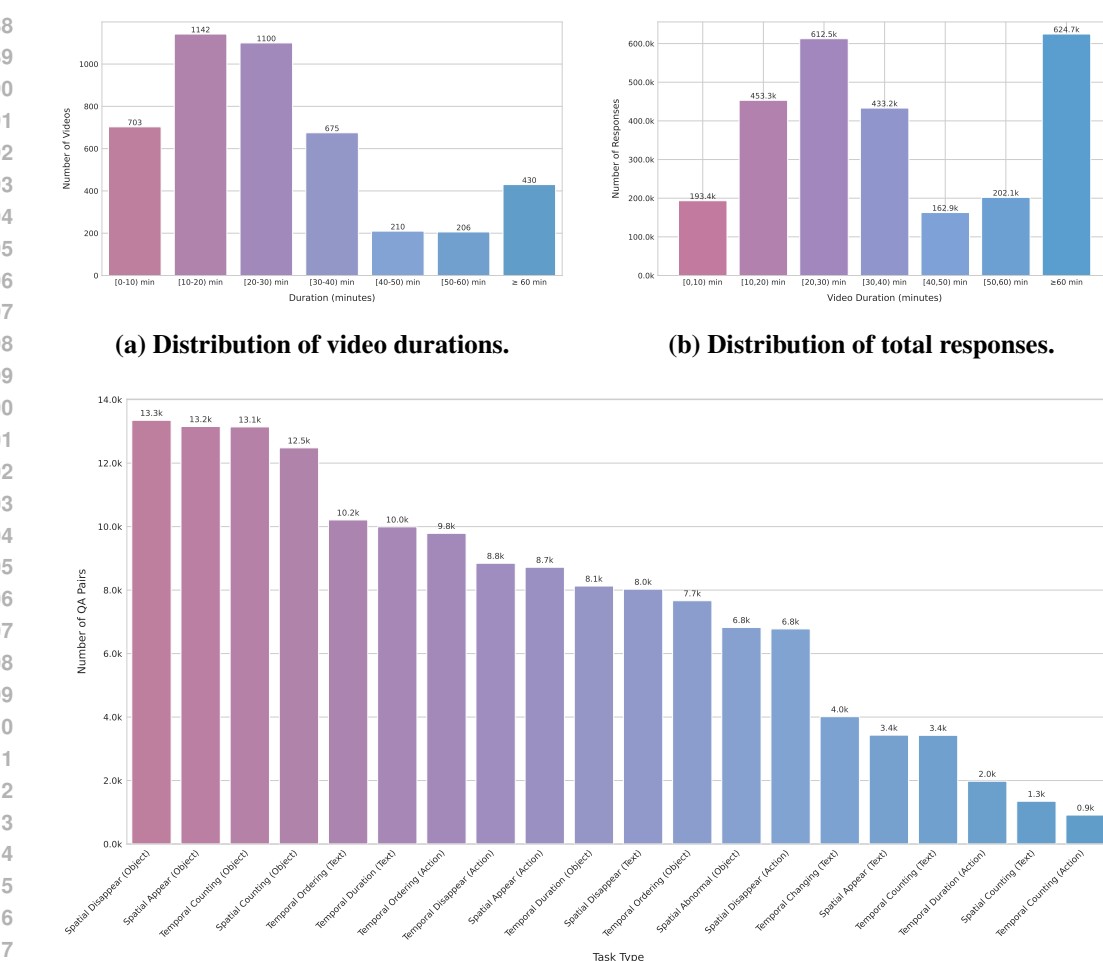

(a) **Distribution of video durations.**

(b) **Distribution of total responses.**

(c) **Number of QA pairs per task type.**

Figure 9: **Statistics of the Memento-54K dataset.** (a) shows the distribution of video durations ; (b) illustrates the number of responses across different duration; and (c) breaks down the total number of QA pairs by task type.

**Text Temporal Duration.** It detects the duration of continuous visibility and the duration of absence between complete text reappearances. We retain texts with the longest visible or invisible duration.

**Text Temporal Counting.** Its goal is to track how many times a complete text appears across the full video timeline. Texts with moderate appearance counts are retained.

**Text Temporal Ordering.** Its goal is to detect ordered sequences of three distinct texts appearing in succession. We retain frequent sequences and report the time intervals between their elements.

**Text Temporal Changing.** Its goal is to detect when a previously complete text block changes into only part of itself in later frames. The assistant alerts the user whenever such a partial form appears, along with the timestamp of the last full appearance.

### A.5.4 DATASET SUMMARY

In summary, Figure 9 presents the statistics of the Memento-54K dataset. As shown in Fig. 9 (a), the dataset covers a wide range of long-video durations, from 5 minutes to over 7 hours. Notably, Fig. 9 (b) shows that videos exceeding 1 hour account for the largest number of response (from assistant) annotations, highlighting the high density of temporal supervision required for streaming long-form

| Task Name | Evaluation Focus |
|-----------|------------------|
| object_temporal_ordering | Focus on overall description, object targets, and their correct sequence. |
| object_spatial_appear | Focus on overall description and whether the object is correctly identified. |
| object_spatial_disappear | Focus on overall description and whether the object disappearance is correctly captured. |
| object_temporal_counting | Focus on overall description, object identity, and correct appearance count. |
| object_spatial_counting | Focus on overall description, object identity, and quantity in the frame. |
| object_spatial_abnormal | Focus on object count and presence of distinct types or variants. |
| object_temporal_duration | Focus on correct object identity and time duration since last occurrence. |
| text_spatial_appear | Focus on correctness of textual content appearance. |
| text_spatial_disappear | Focus on correctness of textual content disappearance. |
| text_spatial_counting | Focus on correct textual content and count. |
| text_temporal_counting | Focus on correct textual content and appearance frequency. |
| text_temporal_duration | Focus on textual identity and elapsed time since disappearance. |
| text_temporal_changing | Focus on content variations and the timing of changes. |
| text_temporal_ordering | Focus on correctness of textual sequence. |
| action_spatial_appear | Focus on correctness of the action appearance. |
| action_spatial_disappear | Focus on the overall disappearance of an event. |
| action_temporal_ordering | Focus on sequence and identity of events. |
| action_temporal_duration | Focus on action correctness and duration. |
| action_temporal_disappear | Focus on correct identification of the action disappearance. |
| action_temporal_counting | Focus on event identity and number of occurrences. |

Table 9: Task-specific evaluation focus used in MementoBench scoring.

video understanding. Fig. 9 (c) further breaks down the QA distribution by task type, demonstrating the coverage across diverse spatial, temporal, and multimodal understanding categories.

In fact, the dataset is hierarchically structured into three levels: (1) **Responses**, each representing an individual assistant reply to an online user query; (2) **QA Pairs**, each composed of a single user question and its corresponding set of responses; (3) **Samples**, formed by randomly selecting 1-5 QA pairs from a single video. These samples comprise the 53,824 final entries in the Memento-54K.

## A.6 MEMENTOBENCH EVALUATION SCORING

As described in the main paper, the *Score* metric in MementoBench is designed to assess the quality of model responses within the temporal window used by *TimeRecall*. To ensure consistency and interpretability, we use GPT-3.5-turbo-0125 as an expert judge with a structured prompt.

---

### Scoring Prompt Used in MementoBench Evaluation

You are an expert evaluator responsible for assessing the **Answer Accuracy** of AI-generated responses based on a given user question (if any), multiple AI outputs, and a reference answer.
Each evaluation is guided by a **Task Name** and its **Key Evaluation Focus**, which indicate the specific goal and assessment priorities for this task. Please review the responses accordingly, with emphasis on the core evaluation points.

*Note: Aspects not included in the core evaluation focus are considered supplementary. These are not required for the response to be deemed correct, but their accurate inclusion may enhance the overall quality.*
*For tasks involving estimates (e.g., time, quantity), a certain degree of deviation is acceptable.*

_______________________________________

**Task Name:** {task_name}
**Key Evaluation Focus:** {task_focus}
**Scoring Criteria (1–10):**
  • **1–2:** Response is incorrect or contains major factual/logical errors.
  • **3–4:** Partially correct, with notable inaccuracies or omissions.
  • **5–6:** Mostly correct, but lacks completeness or task-specific focus.
  • **7–8:** Accurate, detailed, and clearly aligned with the task goals.
  • **9–10:** Fully correct, well-structured, and adds value beyond expectations.
**Instructions:**
  • Assign a score (1–10) to each response.
  • Choose the highest score as the **Overall Score**.
  • Provide a concise explanation, referencing correctness and task alignment.
**Output Format:**

```
{"Overall Score": <score>, "Explanation": "<concise rationale>"}
```

---

Each scoring instance provides the task name, a user query (if available), multiple generated responses, and a reference answer. The judge is guided by a task-specific evaluation focus and is instructed to (1) assign a score between 1 and 10 to each response and (2) provide a concise explanation grounded in task relevance and factual accuracy. The final score is taken as the maximum across all candidate responses.

This explanation-enhanced evaluation enables finer-grained judgment, especially in borderline cases, and improves transparency for evaluation. The task-specific focuses are listed in Table. 9.

## A.7 DECLARATION OF LLM USAGE

Large Language Models (LLMs) were not involved in the conception, design, or implementation of the core methodology in this research. Their usage was limited to assisting with language polishing and improving the clarity of writing. No original scientific contributions, technical innovations, or non-standard components relied on LLMs.