# OpenReview forum: "Memento: Toward an All-Day Proactive Assistant for Ultra-Long Streaming Video"
_ICLR.cc/2026/Conference — ICLR 2026 Poster_

### Official Review · Reviewer_Rvrh · 2025-10-21

**Soundness:** 3
**Presentation:** 3
**Contribution:** 3
**Rating:** 6
**Confidence:** 3

**Summary:**

The paper proposes Memento-54K and MementoBench, a dataset and benchmark for proactive long egocentric video understanding. The paper also presents Memento, a training-based framework for streaming proactive video understanding with multimodal large language models. Memento enables ultra long video understanding by devising a dynamic memory mechanism and an efficient memory retrieval method. Qualitative and quantitative evaluations show that Memento greatly outperforms existing offline or online multimodal large language models in terms of proactive video understanding tasks.

**Strengths:**

- The paper is mostly well written, clear and easy to understand. The problem is well-motivated with clear motivating examples in the beginning demonstrating the failure modes of existing methods.
- The paper presents a large amount of work, including a framework, a dataset and a benchmark, which are all good contributions to the field.
- The paper presents a training-based framework. Despite being more expensive, training-based methods have more potential than training-free methods, and are more technically challenging to get to work. The paper presents many details about their model training strategy such as the step-aware memory attention (SAMA) that could be used in other streaming understanding applications.
- The paper includes detailed ablation studies that provides useful insights for configuration the model and training.

**Weaknesses:**

- The paper only presents a very limited number of baselines in the experiments, mostly just VideoLLM-online, which has been shown to be a fairly weak model even when compared to others in the online setting. I would encourage the authors to add more simple baselines with existing offline video models. More specifically, if would be helpful to show quantitative results for the "long video LLMs" row in figure 1 and further demonstrate the inefficacy of existing models regardless of prompting strategies.
- Although the paper devises a dynamic memory mechanism to prevent undesirable accumulation of visual tokens over time, it still requires repeated filtering of the memory for each user query (line 182). Hence it is a valid concern that the model might not scale very well with a much bigger number of queries (which is completely conceivable in real-world applications). I hope the authors to provide more discussion on this limitation.

**Questions:**

- Are the quality of the LLM-generated content for the dataset and benchmark verified by humans? Given that LLM-generated content can still be unreliable, it is important and common practice to manually inspect and verify the QA-pairs or detected objects/text. It would also be helpful to provide some examples in the appendix and discuss some failure cases, if any.

---

> ### Author Response · Authors · 2025-11-26
> **Rebuttal by Authors**
>
> **W1: Baseline Comparisons**
>
> We appreciate the reviewer’s suggestion to expand the experimental comparisons. We have added two advanced and fully open-source online video models as additional baselines: **[TimeChat-Online (ACM MM 2025)](https://github.com/yaolinli/TimeChat-Online)** and **[Dispider (CVPR 2025)](https://github.com/Mark12Ding/Dispider)**. These represent the strongest existing systems designed specifically for online and proactive video understanding.
>
> ### Table R1. Evaluation with other SOTA models on MementoBench
>
> | Method    | TimeRecall ↑ (Sp.) | TimeRecall ↑ (Temp.) | TimeRecall ↑ (Long >25min) | TimeRecall ↑ (Avg.) | Score ↑ (Sp.) | Score ↑ (Temp.) | Score ↑ (Avg.) | Redund. ↓ |
> |----------|--------|--------|---------|-----------|---------|-------|-------|----|
> | Dispider-offline   | 5.3% | 11.2% | 6.7%      | 7.5%    | 0.53 | 0.31   | 0.48    | -  |
> | VideoLLM-online   | 6.1% | 11.8% | 0.1%     | 8.1%    | 1.55    | 1.21  | 1.40  | 56.4%  |
> | VideoLLM-online*   | 7.9% | 11.6% | 0.3%      | 8.9%  | 5.11  | 5.68    | 5.32   | 21.3%   |
> | Timechat-online   | 10.6%  | 15.7%  | 1.2%   | 11.2%   | 2.08 | 2.67  | 2.53   | 44.6%   |
> | Timechat-online*   | 17.3% | 23.1%    | 1.6%  | 18.3%    | 4.09  | 4.48   | 4.27   | 57.8%   |
> | **Memento**   | **45.9%**  | **51.3%**    | **35.2%** | **47.5%**      | 4.31 | 4.02  | 4.22     | 64.5%     |
>
> However, the public release of Dispider only supports an offline-only mode, and this limitation directly reveals why offline long-video LLMs fail under proactive streaming requirements. Since offline models require the entire relevant temporal segment to be provided at once, they fundamentally lack proactive responding capability. As demonstrated in Table R1, offline systems simply do not trigger at the correct moment, leading to extremely low TimeRecall. The only alternative is to manually query the model at every potential timestamp until the answer emerges, effectively simulating streaming by repeatedly prompting the offline model. As mentioned in both [VideoLLM-online](https://arxiv.org/abs/2406.11816) and Dispider, this strategy incurs prohibitively high inference latency, making it incompatible with real-time or proactive interaction. Therefore, these methods cannot serve as a meaningful baseline for true streaming video proactive reasoning.
>
> **W2: On Scalability of Memory Filtering**
>
> We thank the reviewer for the comment. The selection step in QMS is applied once over all queries jointly, based on the merged memory produced by the dynamic memory module. It is not re-executed independently for each query. As a result, increasing the number of queries does not introduce repeated filtering nor additional memory overhead, as the model always operates over the processed memory and QMS performs only a single lightweight relevance computation. Importantly, QMS does not modify the memory nor restrict it to only a tiny number, ensuring stable performance even when the number of queries increases.
>
> **Q1: On Verification of LLM-Generated Dataset Content**
>
> We appreciate the reviewer’s emphasis on dataset reliability. While the initial QA pairs and annotations in Memento-54K were generated by a combination of GPT-4o, Qwen2-VL, ChatReX, and rule-based pipelines, all generated content underwent an extensive phase of human validation. As noted in Section 4.1, any failed or invalid cases were manually corrected. Specifically, to ensure the dataset’s usability and accuracy, we conducted a full manual verification pass over all 53.6k QA samples. This process identified 10,543 samples (approximately 20%) that required correction. Among these corrected samples:
>
> - **71.2% (≈7,500 samples)** involved timestamp misalignments.
>
> - **57.6% (≈6,000 samples)** involved hallucinated content (describing non-existent objects, texts, or actions).
>
> - **42.4% (≈4,470 samples)** included object, text, or action naming inconsistencies or missing answers.
>
> - **18.8% (≈1,980 samples)** required rewriting of the uninformative questions that may cause vague answers.
>
> More than **70%** of the corrected samples involved multiple types of issues.
>
> We will further provide representative examples and corrected cases in the appendix. We thank the reviewer again for highlighting the importance of transparent data validation.

---

### Official Review · Reviewer_xktG · 2025-10-23

**Soundness:** 3
**Presentation:** 4
**Contribution:** 4
**Rating:** 4
**Confidence:** 3

**Summary:**

This paper introduces Memento, a framework that aims to endow video-language models with persistent long-term memory for streaming video understanding. Instead of treating each clip independently, Memento incrementally updates a memory bank of “key events” or “summaries,” enabling the model to maintain temporal continuity and answer long-term queries across extended video streams.

**Strengths:**

* The paper tackles an important and timely problem, i.e., enabling continuous, long-duration video understanding rather than static or episodic reasoning.
* The conceptual framing (“all-day memory”) is inspiring and forward-looking.
* The hierarchical structure separating short-term vs. long-term memory mirrors cognitive and neuroscience-inspired models, offering conceptual richness.
* Memento appears flexible enough to plug into existing Video-LLMs, potentially generalizing beyond the specific tested backbone (LLaMA 3.1).
* The paper presents detailed ablation studies.

**Weaknesses:**

* It’s not fully evident why such a complex hierarchical design is necessary.
* The overall design reads as over-engineered for the stated goal. Similar effects might be achievable with simpler keyframe/event summarization or query-conditioned caching.
* Experiments are restricted mainly to OVO, with no results on other benchmarks (e.g., StreamingBench, EgoSchema, LongVideoBench, etc.), making the generality claims weak.
* Comparisons are missing with more baselines (for instance non-streaming methods).
* No offline results or cross-dataset evaluations are provided. It’s unclear if Memento benefits standard (non-streaming) settings.

**Questions:**

* Why are results shown only on VideoLLM-Online? Have you tested on other long-video or streaming benchmarks such as StreamingBench, LVBench, or EgoSchema?
* Does Memento improve performance in non-streaming (offline) scenarios as well, or is its advantage restricted to streaming conditions?
* The paper emphasizes long-term efficiency — could you provide actual latency, memory usage, or computational cost metrics to substantiate this claim?
* How does Memento compare to other recent memory-based systems under similar settings?

**Details Of Ethics Concerns:**

I do not identify any significant ethical issues in this paper. The method operates on publicly available video datasets commonly used in the community, and there is no indication of privacy violations, harmful content generation, or misuse potential beyond standard concerns in visual understanding research. Therefore, I do not see any ethical concerns requiring further attention.

---

> ### Author Response · Authors · 2025-11-26
> **Rebuttal by Authors (1/2)**
>
> **W1/W2: On the Necessity of Our Mememto Design**
>
> We thank the reviewer for the comment. We would like to emphasize that the problem studied in this work is inherently very challenging. Handling **ultra-long** videos, **in an online streaming manner**, while also supporting **proactive response triggering**, requires the model to manage long-horizon temporal information, operate without future context, and decide when to answer in real time. **These three challenges interact tightly, and in practice we found that seemingly simple designs struggle to meet them simultaneously.**
>
> More concretely, approaches such as keyframe selection or query-conditioned caching do not avoid the underlying difficulties. Both are still forms of compression, and therefore face the same challenges as any memory mechanism: how to support proactive interaction training, deal with memory that changes over time, and preserve consistent attention behavior between training and inference. These problems remain the same regardless of how the compression is done.
>
> Event summarization, on the other hand, collapses visual evidence into long text sequences. This only shifts the bottleneck from visual tokens to language tokens: the model must now maintain, condition, and update very long textual histories. More critically, a proactive systems with long text contexts still requires SAMA to maintain training–inference alignment; without it, the model collapses due to inconsistent memory layouts. Thus, summarization does not reduce complexity, but it reintroduces the same fundamental issues while losing visual detail.
>
> Furthermore, prompting-based alternatives suffer from severe latency. To proactively detect events, a prompt-based system must query at every potential moment, and the resulting delays make it incompatible with real-time streaming.
>
> Overall, our Memento is designed to jointly address long-horizon retention, online streaming constraints, and proactive output timing. To the best of our knowledge, no existing “lightweight” method can achieve these goals simultaneously. We believe this goes beyond an engineering combination of components, it represents a step toward addressing a challenging and relatively unexplored problem.
>
> **W3/Q1: Results Beyond OVBench**
>
> To address the reviewer’s concern about generality, we additionally evaluated Memento on StreamingBench, which is explicitly designed for out-of-distribution proactive video understanding. (**Other long-video benchmarks, such as EgoSchema, LongVideoBench and LVBench, do not provide a proactive evaluation protocol**)
>
> ### Table R1. Zero-shot Evaluation on StreamingBench
>  | Model               |       Proactive Output     |
> |--------------------------|----------------------|
> | Human             | **100** |
> | GPT-4o             | 56.86 |
> | Claude 3.5 Sonnet   | **64.71** |
> | LLaVA-OneVision     | 29.55 |
> | Qwen2-VL           | 22.73 |
> | LLaVA-NeXT-Video  | 18.18 |
> | InternVL-V2       | **40.91** |
> | Kangaroo           | 16.00 |
> | LongVA            | 15.91 |
> | VILA-1.5          | 17.65 |
> | Video-LLaMA2      | 0.00  |
> | ⭐️ Flash-VStream      | 1.96  |
> | ⭐️  VideoLLM-online     | 3.92  |
> | ⭐️   Memento   | **24.00**  |
>
> It is also worth explaining why offline models can be evaluated in this setting. Since they cannot respond proactively, they are manually prompted near the answer time to determine whether a response needs to be generated. This gives them a clear advantage, as they only answer when it is known that a response is needed. In addition, the benchmark only counts answered cases as denominator, and does not penalize missing responses. So even if a model skips many queries, it can still achieve a high score. In contrast, our model answers every query proactively, without knowing whether an answer is expected, which is more challenging.
> Without any fine-tuning and despite being trained solely on Ego4D, Memento achieved 24.00% accuracy, substantially outperforming prior online models such as Flash-VStream (1.96%) and VideoLLM-online (3.92%). This demonstrates that our memory mechanism generalizes beyond templated instructions and across domains including TV shows, sports broadcasts, and other third-person videos.

---

> > ### Author Response · Authors · 2025-11-26
> > **Rebuttal by Authors (2/2)**
> >
> > **W4/Q1: Baseline Comparisons**
> > We thank the reviewer for highlighting the need for more baselines. In addition to VideoLLM-online, we have incorporated two advanced and fully open-sourced online models to provide a more complete comparison on MementoBench: **[TimeChat-Online (ACM MM 2025)](https://github.com/yaolinli/TimeChat-Online)** and **[Dispider (CVPR 2025)](https://github.com/Mark12Ding/Dispider)**. These models represent the strongest publicly available systems designed for online or streaming-style video understanding, and thus offer a fair and realistic basis for evaluation.
> >
> > ### Table R2. Evaluation with other SOTA models on MementoBench
> >
> > | Method    | TimeRecall ↑ (Sp.) | TimeRecall ↑ (Temp.) | TimeRecall ↑ (Long >25min) | TimeRecall ↑ (Avg.) | Score ↑ (Sp.) | Score ↑ (Temp.) | Score ↑ (Avg.) | Redund. ↓ |
> > |----------|--------|--------|---------|-----------|---------|-------|-------|----|
> > | Dispider-offline   | 5.3% | 11.2% | 6.7%      | 7.5%    | 0.53 | 0.31   | 0.48    | -  |
> > | VideoLLM-online   | 6.1% | 11.8% | 0.1%     | 8.1%    | 1.55    | 1.21  | 1.40  | 56.4%  |
> > | VideoLLM-online*   | 7.9% | 11.6% | 0.3%      | 8.9%  | 5.11  | 5.68    | 5.32   | 21.3%   |
> > | Timechat-online   | 10.6%  | 15.7%  | 1.2%   | 11.2%   | 2.08 | 2.67  | 2.53   | 44.6%   |
> > | Timechat-online*   | 17.3% | 23.1%    | 1.6%  | 18.3%    | 4.09  | 4.48   | 4.27   | 57.8%   |
> > | **Memento**   | **45.9%**  | **51.3%**    | **35.2%** | **47.5%**      | 4.31 | 4.02  | 4.22     | 64.5%     |
> >
> > The results in Table R2 show that Memento significantly outperforms all open-source online baselines across short, temporal, and ultra-long (>25min) proactive settings. This further supports the effectiveness of our memory mechanism for long-horizon proactive reasoning.
> >
> > Moreover, although the paper states that Dispider supports online proactive interaction, its public implementation does not include any online inference pipeline. As a result, we were compelled to evaluate it using the offline execution mode. Unsurprisingly, this leads to extremely low TimeRecall, further reinforcing a key point of this work: **non-streaming long-video LLMs are fundamentally unable to support proactive streaming tasks.**
> >
> > **W5/Q2: Offline Performance**
> >
> > We appreciate the reviewer’s question, but offline performance is not meaningful for our setting. The entire problem studied in this work is online proactive streaming, where the model must decide when to respond while continuously receiving frames. Offline tasks do not face the challenges of temporal triggering or memory stability because they are allowed to access the entire video beforehand.
> >
> > Nevertheless, we note that OVBench also includes offline-style tasks such as Past Memory (PM). Although this is not the focus of our work, Memento still shows competitive performance there. For example in Table 7 of manuscript, on the AR and TR subtasks, Memento outperforms strong offline models such as InternVL2-4B, indicating that our memory mechanism remains effective even when evaluated in non-streaming conditions.
> >
> > In fact, the results in Table R1 and Table R2 show that proactive streaming reasoning is a fundamentally different problem that offline models cannot solve, and our architecture is tailored specifically for this setting.
> >
> > **Q3: Memory Usage**
> >
> > We agree that efficiency claims must be supported by concrete metrics. As shown in Figure 5 (in the main paper), Memento maintains stable memory usage throughout a 4+ hour streaming video, with a peak of only ~45 GB, whereas VideoLLM-online rapidly accumulates visual tokens and runs out of memory with ~80GB within the first hour. This directly demonstrates that our dynamic memory mechanism enables long-horizon streaming to remain feasible, while existing approaches cannot sustain continuous response without exceeding GPU limits.
> >
> > **Q4: Comparison with Other Memory-Based Systems**
> >
> > It is important to clarify that most existing proactive-streaming systems do not use memory mechanism. While among memory-based systems designed for long-video understanding (e.g., MovieChat on OVBench), Memento achieves substantially higher performance under the same evaluation protocol. This demonstrates that our memory mechanism offers clear advantages.

---

> ### Comment · Reviewer_xktG · 2025-11-26
> **Official Comment by Reviewer xktG**
>
> The authors addressed my main concerns. I'm thus increasing my score.

---

### Official Review · Reviewer_kAWu · 2025-10-31

**Soundness:** 3
**Presentation:** 3
**Contribution:** 3
**Rating:** 8
**Confidence:** 4

**Summary:**

The paper presents a vision-language framework that can handle proactive-understanding in a long (multi-hour) streaming video setting. It addressed the memory exploision issues in prior video-LLMs and related work videoLLM-online.

The method mainly provides three modules as the technical contributions:
* It used a dynamic memory module to handle redundant frames, using the heuristic attention based mechnism to compare the similarity of new frames to history frames.
* It introduce a query-related memory selection mechanism to retrieve key frame memory related to current query.
* To handle the frame alignment with the presence of two modules above, it introduces a step-aware memory attention mask to prevent access to expired frames during training.

To evaluate, the paper presents a Memento-54K, a benchmark dataset built on top of existing ego4D dataset, and they generate QA pairs categorized into action, object and text modalities. The ablations confirm the effect of the method. Compared to prior seminar work videoLLM-online (trained similarly), the proposed method can effectively handle multi-hour long inference, while videoLLM-online will be run into OOM after roughly 20 mins in the same setting.

**Strengths:**

Overall I found this paper is well-motivated, technically sound, and validated well with the experiments.
* The proposed three-modules handles the redundant frames, query-memory attention, while still being frame-aligned as videoLLM-online. The three modules compound to each other well. I believe such mechanism can also be extended to other long-video understanding problems.
* The performance on long-video understanding is validated well, on the new proposed dataset, compared in an ablation setting and with prior work. The method in particular highlights its strong recall, which is an impressive improvement.

**Weaknesses:**

* It is not entirely clear how the method will perform in a non-templated QA scenario (if not categorized in the action, object, text query). I am particular interested in that how will the query-related memory selection module generalized in non-templated scenario. It may or may not generalize depends on the complex of the query, which may provide some clues whether current simple cross-attention heuristics is sufficient to handle a more complex setting.

* A small writing issue, the paper use big "M" and small "m" to describe the accumulated historical attentions frames and individual frames if I understand correctly. It will be more easily understood if can be clarified in the writing.

**Questions:**

Mostly the first question I left in the weakness. I think even some simple experiments to evaluate the current model with existing trained checkpoints on a few small cross-domain datasets will be good to give some clues. I would expect it may degenerate, but I am interested to see how reliable it is to handle things if query is getting out of distributions.

---

> ### Author Response · Authors · 2025-11-26
> **Rebuttal by Authors**
>
> We sincerely thank the reviewer for their encouraging and detailed feedback, and we are committed to further improving both the clarity and generality of our work in the final version.
>
> **W1/Q1: On generalization to non-templated or out-of-distribution dataset**
>
> We fully agree with the reviewer that evaluating how the memory mechanism of our Memento generalizes beyond predefined QA templates and across domains is an important direction. To directly assess this, we conducted a small-scale zero-shot experiment on the proactive output task of [StreamingBench](https://arxiv.org/abs/2411.03628), which evaluates proactive video understanding in out-of-distribution settings.
>
> The subset we used consists of 50 samples, each containing one user instruction and one model response. Our model trained solely on first-person daily life videos from Ego4D, was evaluated without any fine-tuning on this benchmark, which features third-person video domains such as TV shows, poker matches, and sports events, as well as instruction formats never seen during training.
>
> Despite the significant domain and instruction gap, Memento achieved 24.00% accuracy, substantially outperforming prior online models such as Flash-VStream (1.96%) and VideoLLM-online (3.92%):
>
> ### Table R1. Zero-shot Evaluation on StreamingBench
>  | Model               |       Proactive Output     |
> |--------------------------|----------------------|
> | Human             | **100** |
> | GPT-4o             | 56.86 |
> | Claude 3.5 Sonnet   | **64.71** |
> | LLaVA-OneVision     | 29.55 |
> | Qwen2-VL           | 22.73 |
> | LLaVA-NeXT-Video  | 18.18 |
> | InternVL-V2       | **40.91** |
> | Kangaroo           | 16.00 |
> | LongVA            | 15.91 |
> | VILA-1.5          | 17.65 |
> | Video-LLaMA2      | 0.00  |
> | ⭐️ Flash-VStream      | 1.96  |
> | ⭐️  VideoLLM-online     | 3.92  |
> | ⭐️   Memento   | **24.00**  |
>
> It is also worth explaining why offline models can be evaluated in this setting. Since they cannot respond proactively, they are manually prompted near the answer time to determine whether a response needs to be generated. This gives them a clear advantage, as they only answer when it is known that a response is needed. In addition, the benchmark only counts answered cases as denominator, and does not penalize missing responses. So even if a model skips many queries, it can still achieve a high score. In contrast, our model answers every query proactively, without knowing whether an answer is expected, which is more challenging.
>
> Still, our model demonstrates strong semantic generalization. For example, in a case from an NBA basketball broadcast ([sample_13](https://huggingface.co/datasets/mjuicem/StreamingBench/tree/main) from StreamingBench):
>
> ```
> [Question (611s)] When USA reaches a score of 95, output "95".
> [Model Output (617s)] The score is 95.
> [Ground Truth (618s)] 95
> ```
>
> Our model successfully detects the event and emits a semantically correct response, despite domain and instruction shifts. This shows that the query-related memory selection mechanism does generalize beyond predefined content and domain, and we expect further gains with cross-domain supervision or instruction tuning.
>
> **W2: On notation clarity**
>
> Thank you for pointing this out. You are absolutely correct. We agree this notation can be confusing without clarification, and will revise the paper to make this distinction clear and consistent in the final version.

---

### Official Review · Reviewer_BySZ · 2025-11-03

**Soundness:** 2
**Presentation:** 3
**Contribution:** 3
**Rating:** 4
**Confidence:** 3

**Summary:**

The paper introduces Memento, a vision-language framework for proactive assistance on ultra-long streaming videos. It proposes a Dynamic Memory (DM) mechanism to fuse redundant frames and a Query-related Memory Selection (QMS) module for efficient retrieval. This paper also introduces Memento-54k and MementoBench, a new benchmark derived from Ego4D for training and evaluating long-term proactive tasks.

**Strengths:**

- The paper addresses a highly significant and practical problem on long-term online video understanding, which has impact on real-life applications.
- The paper is well-written with high-quality figures, making it easy to follow.
- The new benchmark can benefit research in ultra-long streaming videos

**Weaknesses:**

- The only baseline compared is VideoLLM-online. Figure5 shows this model is fundamentally unequipped for the task, hitting OOM at 25 minutes. To strengthen the experiments, the model needs to be compared with other sota models, such as VideoLLM-MoD and LION-FS. To provide a comprehensive evaluation, it would be helpful to validate the proposed method on the Ego4D Narration task that is the same as used in VideoLLM-online.
- Some evaluation metrics are ambiguous. (i) Score is a numeric value from 1 to 10 assigned by gpt3.5. It's unclear if it is a calibrated metric and how good gpt3.5's judegement is. (ii) Redundancy is "the proportion of model responses outside the time window in TimRecall". It's unclear if it's a delay of responses or hallucinations.
- The Dynamic Memory mechanism appears very sensitive to hyperparameters. For example, Figure 6 and Table 4 show that changing $\epsilon$ from 0.7 to 0.8 causes the memory bank to grow 10x larger for only a marginal performance gain. This suggests that the model needs to be carefully tuned when applying to new domains.

**Questions:**

- To solidify SAMA as a key contribution, did the authors attempt to train the DM module without SAMA (i.e., using standard causal attention)? What was the result?
- In fugure6, why does timerecall drop when increasing r_qms from 50% to above 90%? Does this  suggest the LLM gets "distracted" by too much context?
- How to choose the best model, considering the tradeoff between timerecall, score and redundancy?
- How is u in eq3 chosen?

---

> ### Author Response · Authors · 2025-11-26
> **Rebuttal by Authors (1/3)**
>
> **W1: Baseline Comparisons**
>
> We thank the reviewer for pointing out additional baselines. Because the provided link in [VideoLLM-MoD](https://github.com/showlab/videollm-online) redirects to the VideoLLM-online repository and the official repository of [LION-FS](https://github.com/JiuTian-VL/LION-FS) contains only documentation, we select two advanced and fully open-sourced online methods as fair and realistic baselines: **[TimeChat-Online (ACM MM 2025)](https://github.com/yaolinli/TimeChat-Online)** and **[Dispider (CVPR 2025)](https://github.com/Mark12Ding/Dispider)**.
>
> ### Table R1. Evaluation with other SOTA models on MementoBench
>
> | Method    | TimeRecall ↑ (Sp.) | TimeRecall ↑ (Temp.) | TimeRecall ↑ (Long >25min) | TimeRecall ↑ (Avg.) | Score ↑ (Sp.) | Score ↑ (Temp.) | Score ↑ (Avg.) | Redund. ↓ |
> |----------|--------|--------|---------|-----------|---------|-------|-------|----|
> | Dispider-offline   | 5.3% | 11.2% | 6.7%      | 7.5%    | 0.53 | 0.31   | 0.48    | -  |
> | VideoLLM-online  | 6.1% | 11.8% | 0.1%     | 8.1%    | 1.55    | 1.21  | 1.40  | 56.4%  |
> | VideoLLM-online*   | 7.9% | 11.6% | 0.3%      | 8.9%  | 5.11  | 5.68    | 5.32   | 21.3%   |
> | Timechat-online   | 10.6%  | 15.7%  | 1.2%   | 11.2%   | 2.08 | 2.67  | 2.53   | 44.6%   |
> | Timechat-online*   | 17.3% | 23.1%    | 1.6%  | 18.3%    | 4.09  | 4.48   | 4.27   | 57.8%   |
> | **Memento**   | **45.9%**  | **51.3%**    | **35.2%** | **47.5%**      | 4.31 | 4.02  | 4.22     | 64.5%     |
>
> In particular, **Dispider-offline** requires special clarification: although the paper states that it supports online proactive interaction, the released code does not provide related scripts. We therefore were forced to evaluate it using the **offline inference pipeline**, which naturally results in extremely low TimeRecall and poor proactive behavior. This further indicates that offline long video LLMs are fundamentally unsuitable for streaming proactive tasks.
>
> Furthermore, regarding **Ego4D Narration**, this evaluation protocol is fundamentally incompatible with memory-based online models. The evaluation used in VideoLLM-online performs parallel next-token prediction (NTP) over the entire sequence, requiring the model to simultaneously process all tokens. Due to this non-streaming evaluation setup, a memory-based method cannot incrementally feed frames. Instead, all past memory states must be concatenated and processed in a single forward pass, which results in infeasible computational costs.
>
> To provide a fair and practically meaningful comparison, we therefore use **OVBench** explicitly designed for online streaming video understanding. Importantly, this benchmark already includes VideoLLM-online as a built-in baseline, along with many other open-source models.
>
> ### Table R2. Performance comparison on OVBench
> | Method  | AVG  | AA   | GSP  | MP   | AP   | SV   | OP (THV)   | AR   | PR   | TR   | AL   | OP (SP)  | AT   | OT   | AS   | SL   | OES  |
> |:--------:|:------:|:------:|:------:|:------:|:------:|:------:|:------:|:------:|:------:|:------:|:------:|:------:|:------:|:------:|:------:|:------:|:------:|
> | ⭐ VideoChat-Online   | **54.9** | 64.1 | 59.7 | 16.6 | 63.1 | **58.3** | **62.8** | **42.2** | 54.4 | **70.6** | **54.1** | 24.8 | **88.7** | **48.5** | 73.0 | 25.9 | **71.7**|
> |LLaVA-OneVision   | 49.5 | **68.0** | **62.7** | **35.9** | 58.4 | 50.3 | 46.5 | 29.4 | 60.7 | 58.0 | 43.1 | 14.2 | 86.5 | 49.7 | 70.7 | 28.1 | 30.2 |
> | InternVL2        | 48.7 | 52.6 | 60.2 | 27.6 | 57.5 | 52.0 | 58.5 | 38.8 | **67.1** | 58.3 | 38.1 | 31.3 | 87.4 | 37.0 | 75.4 | 31.4 | 5.9  |
> | LongVA    |  43.6 | 64.1 | 56.5 | 29.5 | 54.9 | 51.9 | 34.8 | 35.3 | 55.6 | 57.7 | 31.6 | 3.4  | 67.4 | 44.7 | **80.0** | 26.7 | 4.0  |
> | LLaMA-VID   |  41.9 | 43.6 | 50.9 | 19.6 | **64.0** | 47.5 | 46.8 | 29.4 | 48.9 | 51.2 | 31.9 | 11.2 | 75.7 | 24.8 | 59.1 | 26.0 | 40.0 |
> |⭐ Flash-Vstream   |  31.2 | 26.9 | 37.6 | 23.9 | 60.1 | 41.9 | 40.0 | 23.4 | 35.3 | 26.1 | 24.7 | 28.8 | 27.0 | 21.4 | 29.8 | 25.6 | 26.8 |
> | ⭐ MovieChat     |  30.9 | 23.1 | 27.5 | 23.6 | 58.4 | 43.9 | 40.3 | 25.6 | 31.1 | 23.9 | 26.9 | 39.6 | 24.4 | 28.9 | 29.3 | 25.5 | 21.9 |
> | ⭐ VideoLLM-Online  | 9.6  | 0.0  | 1.8  | 20.9 | 5.2  | 5.9  | 32.6 | 0.0  | 2.3  | 26.7 | 0.6  | 26.6 | 0.9  | 19.9 | 0.9  | 1.7  | 8.3  |
> | **⭐ Memento (Ours)**     | 48.5 | 48.7 | 59.9 | 35.6 | 57.9 | 53.7 | 60.5 | 32.2 | 57.6 | 57.4 | 36.3 | **40.1** | 64.8 | 36.5 | 64.9 | **36.8** | 33.6 |
>
> Overall, the results on MementoBench show that Memento is significantly stronger than all existing open-source streaming models, and maintains robust performance across ultra-long and proactive tasks. On OVBench, Despite using only 1/7 of the training data compared to larger baselines, our method achieves strong performance, outperforming long-context models such as LongVA and MovieChat, and matching the performance of InternVL2. Notably, our model surpasses other online understanding methods like Flash-VStream and VideoLLM-online.

---

> > ### Author Response · Authors · 2025-11-26
> > **Rebuttal by Authors (2/3)**
> >
> > **W2: Evaluation Metrics**
> >
> > 1) **Score metric.** We agree that evaluating free-form language output inherently requires a proxy for human judgment. The GPT-evaluation used in our benchmark follows the same practice adopted in widely used free-form video-language understanding such as [CG-Bench (ICLR 2025)](https://openreview.net/forum?id=le4IoZZHy1), [MMBench-Video (NIPS 2024)](https://mmbench-video.github.io/), and many recent LVLM benchmarks. Under a fixed evaluation rubric, GPT-3.5 could provide highly stable and consistent judgments. Moreover, we output both a numeric score and a textual rationale. In our evaluation, these two components are strongly aligned.
> > ```
> > [Model Output] Text content 'detergent' appeared at 484.11216707507507 begins to leave the video screen.
> > [Ground Truth] Text content 'detergent' appeared at 483.5204872881356 begins to leave the video screen.
> > [Score] 9
> > [Explanation] Candidate provides the correct response with the text content 'detergent' appearing at the specified time of 484.11216707507507s, aligned with the groundtruth reference.
> > ```
> > As this example shows, when the model output matches the ground truth, GPT-3.5 assigns a high score and explicitly states the match; when it mismatches, the score is lower and the explanation points out the error. This clear correspondence between matching quality and scoring confirms that the metric reliably captures response correctness.
> >
> > 2) **Redundancy metric.** Our definition includes both delayed and hallucinated responses, because in proactive video assistance, any output that appears at a time it should not appear creates misleading feedback for the user. For example, even a semantically correct answer is seen unnecessary redundancy, if it is produced too late. In fact, this answer is a “false alarm”, which does not help the user act in time.
> >
> > **W3: Dynamic Memory**
> >
> > We clarify that the observed sensitivity reflects a controllable trade-off in memory usage, not instability in model performance. For the purpose of identifying the most cost-effective threshold, we explained that 0.8 brings only marginal gains while dramatically increasing memory usage, and therefore 0.7 is the optimal trade-off.
> >
> > In fact, on the OVBench benchmark, **we reuse the same configuration ($\epsilon$ is 0.7) for experiment, without any tuning**. As shown in Table R2, Memento achieves strong and consistent results, outperforming VideoLLM-online and Flash-VStream, and closely matching InternVL2. This confirms that the threshold generalizes well across domains, and the ablation study fulfills its purpose to identify a cost-effective configuration without sacrificing accuracy.
> >
> > **Q1: On training DM without SAMA**
> >
> > We appreciate the reviewer’s question. However, training the DM module without SAMA is fundamentally infeasible, because standard causal attention intrinsically breaks the alignment between training and streaming inference in memory-based video models.
> >
> > 1) Causal attention assumes a strictly ordered sequence of tokens and enforces a left-to-right temporal structure. This assumption fails in the memory module. If standard causal attention is naively applied over memory, each new frame’s query attends to all prior memory entries, implying that current time "see" all earlier memory slots.
> >
> > 2) This introduces two fundamental problems. **First, it creates semantic interference**, as the attention pattern no longer respects the independence of memory entries. **Second, it breaks training-inference alignment**. To simulate such full-history attention during training, the model must retain both historical and current memory at each step, resulting in significantly longer token sequences than fixed-length methods and making it infeasible to support ultra-long videos. **However, as soon as truncating or sampling the memory, the attention behavior during training is inconsistent with inference.**
> >
> > 3) Therefore, SAMA is specifically designed to resolve these issues by rethinking how memory is attended to: it enables consistent, sparse, and semantically controlled memory access, preserving alignment while keeping computation tractable. **SAMA is not an optional component, it is necessary to making the DM module trainable and functionally correct in the streaming setting.**

---

> > > ### Author Response · Authors · 2025-11-26
> > > **Rebuttal by Authors (3/3)**
> > >
> > > **Q2: QMS Ratio**
> > >
> > > We appreciate the reviewer’s observation, which is indeed correct. When $r_{qms}$ is too high, the model retrieves a large number of memory entries with weakly relevant information.  In such cases, more context can actually hinder performance, as the model may become less focused due to noise from less relevant content.
> > >
> > > **Q3: On Model Selection**
> > >
> > > We appreciate the reviewer’s question. In selecting the best model configuration, we prioritize improvements in TimeRecall, even at the cost of slightly lower Score, which directly reflects the model’s ability to proactively respond to important events in time. More importantly, we believe that response quality can be further improved through more epochs or offline supervised data. For balancing Redundancy, our practical strategy is to ensure that  redundancy grows in a moderate manner when recall improves. We agree, however, that a more principled way to capture this trade-off is valuable. To that end, we introduce a simple F1-style score that combines TimeRecall with Redundancy.
> > >
> > > $
> > > \text{Score} = \frac{2 \cdot \text{TimeRecall} \cdot (1 - \text{Redundancy})}{\text{TimeRecall} + (1 - \text{Redundancy})}
> > > $
> > >
> > > Taking Table 4 in the main paper as an example, this score could provide a unified measure of how well a model balances proactive coverage and response precision. The results are shown below:
> > >
> > > ### Table R3. Ablation on Memory Schema with F1-style Score
> > >
> > > | Memory Schema | TimeRecall | Redundancy | F1-style Score | Score |
> > > |---------------------|------------|----------------|------------|----------------|
> > > |  $\epsilon=0.6$  | 23.8%       | **50.9%**          |    32.1%       | **5.16**          |
> > > |  $\epsilon=0.7$  | 40.4%       | 56.2%         |   **42.1%**        | 4.39          |
> > > |  $\epsilon=0.8$   | **44.7%**       | 61.4%          | 41.4%          | 4.43         |
> > >
> > > This F1-style score captures the overall effectiveness under the trade-off between recall and redundancy, which can serve as a practical and interpretable indicator for selecting optimal configurations under constrained conditions. We will include this F1-style score in the final version for more comprehensive reporting.
> > >
> > > **Q4: On the Choice of 𝑢 in Eq.3**
> > >
> > > We follow the setting used in [Flash-VStream](https://github.com/IVGSZ/Flash-VStream)’s attention merging module and adopt 𝑢 = 0.2, which was found to provide good information integration in practice. This value is also consistent with the hyperparameter choice in their open-source implementation.

---

> > > > ### Comment · Reviewer_BySZ · 2025-11-27
> > > >
> > > > Thanks for the detailed responses! The additional explanation and experimental results greatly strengthen the paper's soundness. I am inclined to accept this paper and increase my score. I suggest including the newly added experiments in the revision of the paper.

---

### Meta-Review · Area_Chair_DtnY · 2026-01-10

**Summary:**

The paper addresses on highly significant and practical problem on long-term online video understanding problem.

Some concerns are also raised.
1. Insufficient evaluation. missing baseline [Reviewer BySZ, xktG, Rvrh], evaluation metrics are ambiguous [Reviewer BySZ], missing performances in non-templated QA scenario [Reviewer kAWu], experiments restricted mainly to OVO [Reviewer xktG]
2.  Motivation for such technical design is not clear. [Reviewer xktG]
3. Performance sensitive to parameters. [Reviewer BySZ], missing discussion on the limitations [Reviewer Rvrh]


After reading the reviewers' comments and authors' rebuttal, the main concern lies on the insufficient experimental results, which cannot well justify the technical novelties.

The authors have addressed the main concern, and provide more explanations. The problem tackled in significant and practically important.

I am suggesting for acceptance.

**Reviewer Concerns:**

The insufficient evaluations can be well addressed.

**Reviewer Scores:**

The authors addressed the reviewers comments. And I think the reviewers will provide positive opinions on the acceptance of the paper.

---

### Decision · Program_Chairs · 2026-01-26

Accept (Poster)